# A Robust Routing Protocol in Cognitive Unmanned Aerial Vehicular Networks

**DOI:** 10.3390/s24196334

**Published:** 2024-09-30

**Authors:** Anatte Rozario, Ehasan Ahmed, Nafees Mansoor

**Affiliations:** Department of Computer Science and Engineering, University of Liberal Arts Bangladesh (ULAB), Dhaka 1207, Bangladesh; anatte.rozario.cse@ulab.edu.bd (A.R.); ehasan.ahmed.cse@ulab.edu.bd (E.A.)

**Keywords:** unmanned aerial network, flying ad hoc network, cognitive radio network, routing protocol, dynamic topology

## Abstract

The adoption of UAVs in defence and civilian sectors necessitates robust communication networks. This paper presents a routing protocol for Cognitive Radio Unmanned Aerial Vehicles (CR-UAVs) in Flying Ad-hoc Networks (FANETs). The protocol is engineered to optimize route selection by considering crucial parameters such as distance, speed, link quality, and energy consumption. A standout feature is the introduction of the Central Node Resolution Factor (CNRF), which enhances routing decisions. Leveraging the Received Signal Strength Indicator (RSSI) enables accurate distance estimation, crucial for effective routing. Moreover, predictive algorithms are integrated to tackle the challenges posed by high mobility scenarios. Security measures include the identification of malicious nodes, while the protocol ensures resilience by managing multiple routes. Furthermore, it addresses route maintenance and handles link failures efficiently, cluster formation, and re-clustering with joining and leaving new nodes along with the predictive algorithm. Simulation results showcase the protocol’s self-comparison under different packet sizes, particularly in terms of end-to-end delay, throughput, packet delivery ratio, and normalized routing load. However, superior performance compared to existing methods, particularly in terms of throughput and packet transmission delay, underscoring its potential for widespread adoption in both defence and civilian UAV applications.

## 1. Introduction

Unmanned Aerial Vehicles (UAVs) represent groundbreaking technology with diverse applications ranging from defence operations to disaster management and surveillance [1]. These UAVs function within Flying Ad-hoc Networks (FANETs) and can be controlled remotely either manually or through pre-programmed algorithms [2]. Their communication capabilities are facilitated by various modules, such as transceivers and antennas, allowing communication through different technologies like Reliability Factor (RF), satellite, and cellular [3,4].

Establishing network topology within FANETs is critical, and UAVs play a central role in this process. The different topologies, including star, mesh, cluster, and hybrid-mesh, offer varying degrees of connectivity and flexibility [5]. In a star topology, direct connections are established between the ground station and each UAV. Conversely, mesh topology involves a single cluster head linking to the ground station and relaying data among UAVs. Cluster-based topology extends this concept by connecting the ground station to multiple cluster-head UAVs, while hybrid mesh networks allow cluster heads to relay information, creating a dynamic network structure [6].

Communication stands out as the primary method for wireless connections in FANETs, ensuring efficient and reliable communication between UAVs and ground stations. This comprehensive communication infrastructure is essential for enabling the seamless operation of UAVs across various applications and scenarios.

To ensure intelligent connectivity among Unmanned Aerial Vehicles (UAVs), cognitive radio emerges as a valuable tool. Think of cognitive radios as smart devices capable of quickly adapting to different situations. They can make necessary adjustments to parameters like transmitting power, carrier frequency, and modulation strategies based on the current conditions [7]. This adaptability is crucial for optimizing connectivity within the network. When these cognitive radios interact with each other, they form what is known as a cognitive radio network (CRN). It is essentially a sophisticated wireless communication network where the radios can continuously adjust their connectivity parameters to enhance the overall performance of the system [8].

With the rising deployment of Unmanned Aerial Vehicles (UAVs), there is a growing need for streamlined routing and energy-saving strategies [9]. This is crucial to strike a balance between minimizing network delays, ensuring secure data transmission, and upholding data integrity. The primary objective of UAV networks is to exchange of information between UAVs and Ground Stations (GS), relying heavily on efficient routing protocols to facilitate intelligent communication [10]. However, crafting effective routing solutions for UAV networks, particularly in dynamic, densely populated, and energy-sensitive environments, presents notable hurdles [11]. The challenge lies in finding ways to prioritize energy-efficient UAV-to-UAV communication while still upholding Quality of Service (QoS) standards [12].

In this article, a routing protocol for CR-UAVs is presented, treating the problem as a weighted graph issue. It incorporates parameters like distance, speed, direction, link quality, energy, and reliability to optimize route selection, aiming for faster and more reliable data communication. A key feature of the proposed protocol is the introduction of the Central Node Resolution Factor (CNRF). This parameter, influenced by common idle channels and transmitting weight value ($\xi_{T}$), guides routing path decisions. Utilizing the Received Signal Strength Indicator (RSSI) for distance estimation enhances data forwarding and improves network performance in dynamic environments. To address the challenges of high mobility, the protocol employs predictive algorithms to anticipate neighboring nodes’ future locations, effectively preventing link failures and packet drops. Additionally, it defines link quality in terms of delay, encompassing back-off, switching, and queuing delays to estimate link weight, ensuring route stability and reducing link failures. The paper also introduces a robust route selection mechanism to detect malicious nodes or external threats, a clustering strategy, enhancing trust in received messages during data exchange, and a cluster formation mechanism. It outlines route maintenance for managing multiple routes discovered during selection or addressing link failures during data transmission. Simulation results compare the protocol’s performance with itself on different packet sizes and others, focusing on metrics like throughput, communication delay, packet delivery ratio and normalized routing load.

The rest of this work is structured as follows: Section 2 addresses the present protocols for FANETs in brief. This highlighted the advantages and disadvantages of several current protocols. Section 3 examines the proposed routing protocol’s network model. Section 4 goes into detail about the proposed routing protocol. Section 5 route selection and maintenance process for the UAVs to ensure safer connectivity. Section 6 discusses the simulation result. Section 7 finishes the article by discussing the unresolved research questions.

## 2. Related Works

Creating routing protocols for Unmanned Aerial Vehicle (UAV) networks in situations with high mobility and network density poses significant challenges due to the distinctive characteristics of these networks. In this section, we will discuss some of the existing routing protocols employed in UAV networks.

The Reactive-Greedy-Reactive (RGR) protocol utilizes HELLO signals to keep tabs on the whereabouts and identities of nearby nodes. This helps in monitoring the positions and identities of neighboring nodes. The protocol relies on four types of control packets: Route Requests (RREQs), Route Replies (RREPs), Route Error (RERR) messages, and hello signals. While all these packets offer geographical information, they function similarly to the Ad Hoc On-Demand Distance Vector (AODV) protocol [13]. Despite employing a routing table, the RGR protocol is prone to message loss when the closest neighbor is unavailable, leading to the need for frequent path searches. However, it does leverage mobility prediction, which can be advantageous. Nonetheless, this prediction capability might interfere with tracking and search scenarios.

In [14], a refined routing approach that utilizes Optimized Link State Routing (OLSR) is proposed to meet the low-delay requirements within Flying Ad-hoc Networks (FANETs). This scheme aims to minimize the routing overhead stemming from node movements, effectively addressing the challenges presented by the high speed of nodes and frequent updates typical in FANET environments.

The proposed scheme integrates sophisticated OLSR techniques, taking into account various factors such as neighbor connectivity, power consumption, available bandwidth, and employs neural network-based methods for route computation. Through simulations, it is shown to enhance FANET services by reducing control message volume, improving routing efficiency, and achieving superior performance in metrics like packet delivery rate, throughput, and network stability compared to traditional OLSR approaches. The modified OLSR (MOLSR) protocol, which places particular emphasis on ensuring connection stability and optimizing node bandwidth utilization, emerges as a standout performer when compared to the standard OLSR protocol [14].

The CLRM-DSR protocol is designed to improve the efficiency of Dynamic Source Routing (DSR) in Flying Ad Hoc Networks (FANETs) by reducing routing overhead through the evaluation of network quality. Unlike traditional DSR, this protocol tackles issues like increased latency and the absence of endpoint connection information. CLRM-DSR achieves its goal of efficient routing by utilizing cross-layer resource control and connection quality indicators, which are employed in uni-cast routing queries and responses. It excels particularly in scenarios involving mobility and minimizing end-to-end delay. However, it is worth noting that CLRM-DSR may not be the ideal choice for extremely dense networks. This is because it does not fully account for factors like latency, energy efficiency, quality of service, dynamic topology changes, and network dependability [15].

Routing Approach (ICRA) is introduced to tackle the challenges posed by rapidly changing environments. ICRA comprises three key components: the clustering module, the clustering strategy modification module, and the routing module. This approach is tailored for UAV ad hoc networks, which consist of UAV nodes (NU) and a ground station (GS) facilitating communication between UAVs and the ground. It operates under the assumption that nodes are equipped with GPS and employ a network time protocol system for synchronization [16].

ICRA leverages a clustering algorithm to organize UAV nodes into clusters, ensuring a consistent network topology. When compared to fixed-weight approaches, ICRA demonstrates notable advantages across various experimental scenarios. These include faster clustering, enhanced topological stability, improved energy efficiency, and superior quality of service. These improvements are largely attributed to ICRA’s unique utility computation method [16].

The Secure and Reliable Inter-cluster Protocol (SecRIP) aims to improve both Quality of Service (QoS) and Quality of Experience (QoE) metrics by incorporating innovative approaches like chaotic algae and dragonfly methods for managing clusters and facilitating inter-cluster communication. SecRIP operates in two main stages: cluster formation and cluster head selection. Initially, nodes are grouped into clusters using the chaotic algae method, which enhances overall network efficiency and reduces data transmission delays. Following this, the dragonfly method is employed to select cluster heads, enabling efficient message exchange within and between clusters.

While SecRIP demonstrates notable strengths in terms of reliability, energy efficiency, mobility support, minimal end-to-end delays, and high throughput, it does introduce higher latency due to its clustering technique based on the Algae strategy. Additionally, the consideration of network density and dynamic topology may contribute to this increased latency [17].

The CORF protocol [18] a total of two parts in this process: route creation and collision avoidance. The transmitting UAV calculates the path and chooses a relay node based on transmission probability in the first stage. Dijkstra’s Method may determine the shortest path. To avoid collisions with other UAVs, the UAV follows a predefined course in the second stage. For data transfer, the protocol employs Public Information (PI) packets and Acknowledgement (ACK) packets for confirmation. It relies on directional data and transfer probability to select relay nodes, often utilizing the Ground Station (GS). Wi-Fi is used for navigation in the Course Aware Opportunistic Routing protocol [18], which excels in low latency and high mobility. Although it performs well in dynamic topology scenarios, it falls short in several areas, including energy efficiency, quality of service, dependability, throughput, and network.

The Delay and Link Stability Aware (DLSA) routing protocol, detailed [19], is all about ensuring strong and reliable connections. It goes through several stages, starting with initialization, then verification, and finally system segmentation. Here is how it works: DLSA sets up Red-Black priority trees for both aerial and ground networks, with a Cognitive link layer. When it comes to choosing nodes, DLSA gives priority to those that are most crucial for maintaining a stable connection. If a selected node meets certain durability thresholds, data is transmitted directly through it. However, if it does not meet the criteria, DLSA kicks off a route discovery process to find nodes with high durability. Once the right nodes are identified, DLSA picks the channel with the highest durability for cooperative data transmission. This ensures that communication between ground and aerial networks operate smoothly and successfully.

The Gray Wolf Optimizer (GWO) algorithm, inspired by the social structure of gray wolves, is used to improve routing in mobile networks using the GW-COOP (Gray Wolf Algorithm utilizing Cooperative Diversity Technique) routing protocol [20]. It implements collaborative diversity, in which two relays provide dependability assistance for each link. An objective function is used to assess the locations and fitness of nodes, with priority given to the source, $R_{1}$, and $R_{2}$. Critical data is sent immediately, although ordinary data can be sent directly or via relay via $R_{1}$ or $R_{2}$. Enhanced Signal Noise Ratio Combining (ESNRC) is utilized at the destination to choose the best signal among incoming messages based on signal-to-noise ratios. This protocol improves the efficiency and reliability of routing.

The proposed routing protocol overcomes existing limitations by considering factors like residual energy, reliability, and weighted graph values. It aims to address challenges like delays, high bandwidth use, scalability, and low network density for improved energy efficiency and reliability.

The Novel Optimized Link-State Routing Scheme (OLSR+GPSR) is an innovative advancement in Flying Ad-hoc Networks (FANETs). By integrating optimized link-state routing (OLSR) with greedy perimeter stateless routing (GPSR), it introduces several advantages. It employs a fuzzy system to adjust the hello message broadcast period based on UAV velocity and position prediction error, ensuring that high-speed UAVs have shorter broadcast periods. MPR nodes in OLSR+GPSR are determined using multiple metrics such as neighbor degree, node stability, buffer capacity, and residual energy, leading to more stable and efficient routes. The scheme also reduces routing overhead by eliminating two OLSR phases: TC message dissemination and route calculation. Simulations show that OLSR+GPSR outperforms existing methods like P-OLSR, and OLSR-ETX in terms of delay, packet delivery ratio, throughput, and overhead [21].

The paper [22] presents a novel approach for using UAVs as relays to maximize transmission coverage in challenging terrain. It employs a combination of optimization functions, viewshed analysis, the traveling salesman problem (TSP), and the A* search algorithm for efficient path planning. The optimization function focuses on minimizing the cost while ensuring positive results. Viewshed analysis, conducted at 100 m altitude, identifies visible regions and avoids areas with inadequate coverage by generating binary raster data. The TSP is used to determine the optimal sequence for visiting points of interest, minimizing total distance and energy consumption. The A* search algorithm utilizes irregular terrain elevations to calculate the minimum cost path between points. The results highlight the importance of terrain profiles in signal pathloss calculations and demonstrate the effectiveness of the proposed approach in using drones as signal relays. This method ensures efficient UAV relay planning by optimizing coverage and navigation across diverse and challenging terrains (Figure 1).

## 3. Network Model

A Flying Ad hoc Network (FANET) is essentially a dynamic mobile network that possesses unique capabilities allowing direct communication between Unmanned Aerial Vehicles (UAVs), satellites, and ground stations. This setup is crucial for ensuring seamless data transmission to control centers and facilitating collaboration among UAVs, enabling large-scale ad hoc networking across multiple UAVs simultaneously. FANETs are designed to accommodate the inherently high mobility of UAVs, which distinguishes them from other types of networks such as Mobile Ad hoc Networks (MANETs) and Vehicular Ad hoc Networks (VANETs). This mobility factor plays a central role in various applications ranging from defence inspection to disaster response and aerial photography.

A Cognitive Radio Network (CRN) is essentially a smart communication system that can adapt its operations based on the prevailing network conditions. It is like having a network that can think for itself and adjust its processes accordingly. In CRNs, there are two main types of users: Primary Users (PU) and Secondary Users (SU). Primary users hold the licenses for using specific portions of the spectrum, they have exclusive rights to these frequency bands. On the other hand, Secondary Users are unlicensed users who can utilize unused spectrum bands owned by primary users. This concept, pioneered by J. Mitola III, aims to make more efficient use of the radio spectrum by allowing secondary users to access unused spectrum resources [23].

Now, Cognitive Radio Ad-hoc Networks (CRAHNs) are a specific type of CRN that focuses on cluster-based networks. In these networks, nodes are grouped into logical clusters, where all the nodes within a cluster are located close to each other geographically and are interconnected. This clustering approach helps in organizing the network more efficiently and managing communications within the cluster effectively [23].

In the proposed Cognitive Radio UAV networks (CR-UAV), intermediate nodes play a crucial role as they are deemed reliable and are strategically selected as central nodes based on their path weight. These central nodes form logical groups of nodes, with each group having a designated Central Node (CN) responsible for overseeing efficient communication within the group.

Central Nodes (CNs) are pivotal in managing communication between different groups, facilitating seamless inter-group communication. When Primary Users (PU) become active and interfere with the network, the CR-UAV network dynamically reorganizes itself to find alternative routes and maintain connectivity. In cases where nodes become isolated from their groups, they actively seek re-connection by reaching out to new Central Nodes or groups. To facilitate this process, Central Nodes employ a neighbor discovery mechanism. This involves sending out HELLO messages to potential neighbor CNs and awaiting a REPLY. If no response is received within a set time frame, the message is re-transmitted to ensure robust neighbor discovery and network connectivity.

The proposed routing protocol introduced grouping them into logical groups. And every group has its own Central Node considering the node with the highest weight value as the Central Node (CN) and the second-highest as the secondary CN. Central Node (CN) is determined when the group has some networking nodes and needs to be controlled by some other node. CN can make communication between inter-connected nodes also with the intra-connected nodes and other groups’ CN. CN also uses the Nearest Node (NN) to communicate with the other CN’s NN to minimize the transmission energy and for fast communication. NN is used as a communication router between two different network groups, which is considered a common network or common channel. Whenever the PU becomes activated the whole network needs to reform the group because when a PU comes to the network SU needs to find another route for communication as they are using PU’s paid network spectrum.

The clustering mechanism proposed in this article is designed to be independent of any specific Primary User (PU) activity model. For analytical performance evaluation, it adopt the Semi-Markov ON-OFF model, which characterizes PU traffic on any channel. In this model, channels can be in one of two states: busy (ON) or idle (OFF). The duration of these busy and idle periods are treated as independent random variables, reflecting the autonomous operation of PUs who have licensed access to the spectrum bands. Consequently, Secondary Users (SUs) only utilize the available idle channels and must vacate them whenever PU activity is detected. Additionally, it assumes the presence of a global common control channel within the network to facilitate this dynamic spectrum access [24].

The suggested protocol in the CR-UAV network stresses cluster formation, which is critical for improving network performance. A weighted graph algorithm is used to intelligently classify UAVs into clusters depending on their geographical closeness. CNs are nodes with a stronger impact, ensuring that they lead their respective clusters. This strategy improves spectrum use, communication reliability, and data interchange within the CR-UAV network, all of which contribute to the protocol’s overall success.

Central Node selection in the proposed protocol is vital for efficient cluster management within CR-UAV networks. The highest-weighted node within each cluster becomes the Cluster Head (CN), chosen for its strong connectivity and energy reserves. Furthermore, a secondary high-weight node assumes the role of secondary CN (SCN), providing support to the CN and ensuring network resilience. These criteria are central to the protocol’s objective of enhancing network performance and resource utilization in CR-UAV deployments.

The protocol prioritizes robust inter-cluster communication, clustering strong links between Central Nodes (CNs) and neighboring SCN within clusters. This strategic approach aims to facilitate seamless data exchange across clusters, achieved through a sophisticated routing mechanism enabling CNs and SCN to act as data relays when necessary. This enhancement significantly boosts the overall efficiency and performance of the Cognitive Radio UAV network.

Our proposed protocol for Cognitive Radio UAV networks (CR-UAV) relies heavily on efficient cluster management. Each cluster continuously monitors nodes and nearby clusters for any issues, such as the sudden entry of Primary Users (PU), which can disrupt the network. When such issues arise, clusters may need to reform with new Nearest Networks (NN), potentially losing access to previous channels.

To adapt, nodes may discover new channels while traveling, leading to changes in cluster membership. Some nodes may become disconnected and attempt to reconnect with the nearest network cluster or form their own. To mitigate energy depletion or communication interruptions, nodes proactively select new Cluster Heads (CN) or secondary CN (SCN) from high-weight nodes. This proactive approach enhances operational efficiency and network resilience, ensuring success in CR-UAV deployments.

Figure 2 illustrates the overall process for our proposed cluster formation and maintenance protocol in Flying Ad-hoc Networks (FANETs). The flowchart is divided into several stages, each representing a crucial step in the protocol.

Start: The process begins with the initialization of the algorithm.Node Detection and Identification: UAVs scan their surroundings to detect and identify neighboring nodes.Broadcasting and Sharing: Detected nodes broadcast their information, including Adjacency Clustering List (ACL) and Node List (NL), to neighboring nodes.Graph Construction and CNRF Calculation: Each UAV constructs a graph based on the received information and calculates the Cluster Node Relevance Factor (CNRF).Cluster Joining Decision: The UAVs decide whether to join an existing cluster or attempt to form a new one based on the CNRF values. If the decision is No, the UAV attempts to form a new cluster. If the decision is Yes, the process proceeds to the next steps.Condition: CNRF Comparison and Cluster Joining: UAVs compare CNRF values and proceed with the cluster joining process.Cluster Joining Process: Nodes join the appropriate cluster based on the comparison results.Node Move-in Process: Nodes that successfully join a cluster go through the move-in process to integrate into the cluster.Route Discovery: The protocol initiates the route discovery process to establish communication paths. Data Routing and Adjustment: Based on the discovered routes, data routing and necessary adjustments are made. If the data routing is successful (Yes), the protocol continues with data transmission using alternative routes as needed. If the data routing fails (No), the node initiates the move-out process.Node Move-out Process: Nodes that cannot successfully route data will leave the cluster and the protocol ends for those nodes.

## 4. Proposed Routing Protocol

This section outlines the development of the proposed routing protocol, including proposed clustering strategy, neighboring UAV discovery process, cluster formation, central node resolution factor, central node discovery process and delay awareness of the proposed protocol. The reactive protocol considers factors like distance, speed, direction, network quality, and reliability to enhance rapid and dependable data exchange. The proposed routing protocol involves broadcasting Route Request (RREQ) messages to discover routes, with neighboring UAVs replying with Route Reply (RREP) messages containing path delay, reliability factor (RF), and residual energy (RE). UAVs evaluate these replies and select the most reliable paths based on the Central Node Resolution Factor (CNRF), electing Central Nodes (CNs) accordingly. Data packets are routed through these CNs for stability and efficiency, and alternative routes are used if a link fails. The protocol dynamically adjusts to network changes, continuously monitoring link quality and energy levels for optimal performance.

### 4.1. Proposed Clustering Strategy

In our proposed Cognitive Radio UAV networks (CR-UAV), it introduces a distributed clustering scheme aimed at optimizing network performance. This approach adapts channel access timing within each cluster through synchronized super-frames.

In our proposed super-frame setup, the Central Node (CN) initiates the beacon period by broadcasting a beacon message containing essential cluster information such as its ID, time synchronization data, SCH ID, and control/resource allocation details. Following this, during the spectrum sensing period, nodes assess channel availability and update their list of usable channels accordingly. If the channel is clear, the CN proceeds to transmit data; if not, it waits and retries later using a Distributed Coordination Function (DCF). Each cluster has its common channels chosen by the CN, and if a Primary User (PU) appears, the CN switches to the next channel to avoid interference, with other nodes following suit.

In our proposed super-frame setup, the Neighbor Discovery phase immediately follows the Spectrum Sensing period. It has a critical time for both discovering neighbors and forming clusters. Nodes entering the network begin by scanning available channels and compiling a list of accessible channels. They then organize these channels and cycle through them to locate nearby nodes. If a Primary User signal is detected on a channel, nodes move to the next one. The goal is for nodes to spend sufficient time on each channel to identify all nearby nodes. Once neighboring UAVs are found, nodes exchange their lists of available channels and establish connections.

Finally, the data period split into communicating within clusters and communicating between clusters. These phases are very important for making sure our clusters work seamlessly and can communicate with each other effectively in our CR-UAV network (Figure 3).

#### Neighbour Discovery Process

In our proposed network framework, the Neighbor Discovery phase represents a crucial step for Cognitive Radio-UAV (CR-UAV) nodes seeking to join the network. Through this phase, CR-UAV nodes can identify neighboring nodes and/or neighboring clusters, facilitating the exchange of control information essential for network integration.

The process begins with the identification of available free spectrum by the CR-UAV node, followed by the preparation of its Accessible Channels List (ACL). Once the ACL is established, the CR-UAV node arranges the channels in sequential order, optimizing the process for neighbor discovery.

Channel hopping then ensues, as the CR-UAV node sequentially explores the identified channels to listen for beacon signals. To ensure synchronization and coordination, a global common control channel is utilized to obtain a common time reference. During this phase, the CR-UAV node remains on each channel for a defined channel stopover time, allowing ample duration to detect neighboring nodes within the vicinity.

Upon successfully receiving a beacon signal, the CR-UAV node promptly responds by transmitting a HELLO message to the Central Node (CN), initiating communication. Subsequently, the CR node awaits a REPLY message from the neighboring CN. A timer is activated during the transmission of the HELLO message to establish a maximum waiting time for the receipt of the REPLY message.

In cases where no reply is received within the specified time frame or upon failure to receive the REPLY message, the CR-UAV node employs a retransmission mechanism to ensure data transmission reliability. However, it is essential to note that the total duration of all re-transmissions and corresponding timers for any particular HELLO message should not surpass the channel stopover time.

Following successful communication and receipt of a REPLY message from a neighboring cluster, the CR-UAV node proceeds to initiate the node joining process, as detailed further in the Node Move-in and Node Move-out subsection of our paper. Conversely, if no beacon signal is detected from any cluster during the neighbor discovery phase, the CR-UAV node initiates the cluster formation process, a topic explored extensively in the subsequent section of our paper.

Through this comprehensive neighbor discovery process, CR-UAV nodes efficiently establish connections with neighboring nodes and clusters, laying the foundation for seamless network integration and collaboration within the Cognitive Radio environment.

### 4.2. Central Node Resolution Factor

This part of this section introduces the Routing metric Central Node Resolution Factor (CNRF), total cumulative CNRF $\left(N_{P}\right)$ is calculated considering the path P is expressed as follows,
(1)$$N_{P}=\sum_{i,j\in P_{th}\left(s,d\right)}N_{i,j}$$

The source node sends an RREQ to nearby UAVs for data transmission, and neighboring nodes calculate their distance using Received Signal Strength Indication (RSSI) to determine their positions. RSSI is chosen because it is a simple and cost-effective method that can be implemented with existing hardware on commercial wireless devices. While RSSI can vary due to environmental factors such as interference and obstacles, it offers a practical balance between cost and complexity. To mitigate some of the variability and improve accuracy, it employ techniques such as averaging multiple RSSI readings and filtering.
(2)$$d=10^{\frac{k-20log\frac{4\pi l_{0}}{v}}{10\omega }}$$Here, *k* is inspected as path loss, ω is the exponent of signal loss, *v* is the received signal’s wavelength and $l_{0}$ is reference distance. When a UAV receives a signal, it saves its current coordinates and timestamps $T_{1}$. It then waits for a certain amount of time before responding with its coordinates and sending time $T_{2}$. This paper discusses the UAV node’s transmission-related speed (s) may be calculated as follows:(3)$$s=\frac{\sqrt{{\left(\chi_{t}-\chi_{r}\right)}^{2}+{\left(\varphi_{t}-\varphi_{r}\right)}^{2}+{\left(\zeta_{t}-\zeta_{r}\right)}^{2}}}{\left(T_{2}+\Delta \right)-T_{1}}$$Here, the coordinates for UAVs are $\eta_{r}\left(\chi_{r},\varphi_{r},\zeta_{r}\right)$ receiving and sending individually. Δ indicates the transmission time. Consequently, the direction between neighbor and destination nodes is defined during packet reception and transmission.
(4)$$\theta =\cos^{-}1(\frac{\left(\chi_{t}\chi_{r}\right)+\left(\varphi_{t}\varphi_{r}\right)+\left(\zeta_{t}\zeta_{r}\right)}{\left(\sqrt{\chi_{r}^{2}+\varphi_{r}^{2}+\zeta_{r}^{2}}\right)\times \left(\sqrt{(\chi_{s}^{2}+\varphi_{s}^{2}+\zeta_{s}^{2}}\right)})$$The displacement between two nodes in the time of sending packets at the time $T_{2}$ is measured as ($T_{v}$),
(5)$$T_{v}=\int \partial \theta$$So, the weighted value during data transmission $\left(\xi_{T}\right)$ between two nearby nodes is calculated by every other UAV and may shown as follows,
(6)$$\xi_{t}=\frac{\phi_{t}}{T_{V}*\partial_{P}^{L}*s}$$In this case, $\phi_{t}$ represents the data transmission range between two UAV nodes, $T_{v}$ represents the displacement of two UAV nodes, $\partial_{P}^{L}$ is the total of connection delays when routing and *s* is UAV speed during transmission.

The proposed protocol assesses network failure using back-off, switching, and queuing delays. It also handles traffic density issues in congested areas, considering the number of nearby nodes ($UN_{i}$), $DR_{i}$ data rate of $N_{i}$, and *S* as the size of sending packets, so the queuing delay $\partial_{Q}^{D}$ of $N_{i}$ is calculated as follows,
(7)$$\partial_{Q}^{D}=\frac{SU_{i}}{DR_{i}}$$UAV nodes use random back-off time for collision reduction, by using the following equation:(8)$$\partial_{D}^{B}=\frac{1}{\left(\left(1-P_{c}\right)\left(1-\left(1-P_{c}\right)\right)U_{i}-1\right)}W$$Here, $P_{c}$ is symbolized as the probability of collision $UN_{i}$ is the nearby UAV nodes on the channel $CH_{i}$ and *w* is the window size.

Intermediate nodes may switch channels during discovery with minimal time spent, typically considered as zero time. Switching delay $\left(\partial_{D}^{S}\right)$ is the time needed to change channels within a channel group if a drone $UN_{i}$ is required to switch channel *p* to *q* in its channel group whilst forwarding the message to next-hop $UN_{j}$.
(9)$$\partial_{i,j}^{S}=a*\left|p-q\right|$$
where *a* is a positive real number and the tuning delay of two UAVs is regarded to have a specified step size *a*. let’s consider step size 5 MHz, a is 5 ms [24]. So, it calculates The link delay $\left(\partial_{i,j}^{D}\right)$ using the above equation,
(10)$$\partial_{i,g}^{D}=\partial_{i,j}^{S}+\partial_{D}^{B}+\partial_{D}^{Q}$$The suggested protocol uses the RF factor to provide safe data transfer, which may be computed as follows:(11)$$RF=e^{RN}$$RN represents the reported number of suspected nodes in data transmission, indicating the importance of transmitting weight value $\left(\xi_{T}\right)$ and *N*, Cognitive nodes calculate the CNRF value makes increasing stability during transmission as the UAV nodes choose the highest $\left(\xi_{T}\right)$ in the routing path. $C_{i}$, number of common ideal channels, link delay $\left(\partial_{i,j}^{D}\right)$, the reliability factor is RF, Residual Energy is RE, Residual Energy (RE) is calculated using Initial Energy (IE) minus Consumed Energy (EC). EC includes Transmission Energy (TE), which results from the power used for N packet transmissions.
(12)$$CNRF_{i,j}=\frac{\xi_{t}*C_{i}}{RF*\partial_{i,j}^{D}}\times RE$$

### 4.3. Central Node Discovery Process

In the proposed routing Algorithm 1, the route discovery procedure starts with broadcasting the Route Request (RREQ) to its nearby nodes. The nearby nodes then evaluate the trustworthiness of the mediator nodes using the Suspected UAV Node (SQN) value. If the SQN value exceeds 50%, that particular node is declared malicious and flagged with an infinity value, initiating the node exit process for the suspicious UAVs.

The proposed protocol also examines Residual Energy (RE) and Reliability Factor (RF) during the process. Based on the distance, angle, speed ratio, and number of free channels of a node, a destination node calculates the Central Node Resolution Factor (CNRF). RREQ and Route Reply (RREP) messages are exchanged between nodes to discover all possible paths, collectively referred to as the ‘Path’ of the algorithm. The ‘Path’ represents the overall route that data packets will follow, consisting of a sequence of nodes or waypoints. In contrast, a ‘path’ represents a single segment or connection between two nodes within the overall ‘Path’. Both ‘Path’ and ‘path’ are implemented using list data structures, with each element corresponding to a node or waypoint. The source node calculates and stores the CNRF, RE, and RF, ensuring efficient and reliable routing by selecting the most optimal ‘Path’ based on these parameters.

The Cluster Formation Algorithm runs periodically, optimizing energy consumption by processing, sending, and receiving messages at defined intervals. It involves each node ($UN_{i}$) scanning and constructing an Adjacent Cluster List ($ACL_{i}$) and Node List ($NL_{i}$), which are then transmitted to neighboring nodes ($UN_{n}$). Based on gathered information, each node constructs a graph component ($G_{ci}$) from its previous graph information ($G_{pi}$ ). Each node determines its Central Node Route Factor ($CNRF_{i}$) and shares it with its neighbors. The node with the highest CNRF within its neighborhood is selected as the Cluster Node (CN), forming a cluster and selecting Sub-Cluster Nodes (SCN) based on CNRF values. If there are multiple neighbors with higher CNRF values, the node with the smallest ID is chosen. This process continues until clusters are formed with selected CNs and Cluster Members (CMs). Considering the number of neighbors (*n*) and the number of nodes in the network (*N*), the total number of accessible channels is *C*, and a particular node’s accessible channels is *X*. For one node, the route discovery algorithm has a complexity of O(n+δC), where δC represents the accessible channels for that node. For *N* nodes, the algorithm’s complexity becomes O(nlogN+δC). However, when the number of nodes (*N*) in the network is significantly larger than the constant δC, the complexity simplifies to O(nlogN).
**Algorithm 1:** Central Node Discovery Process
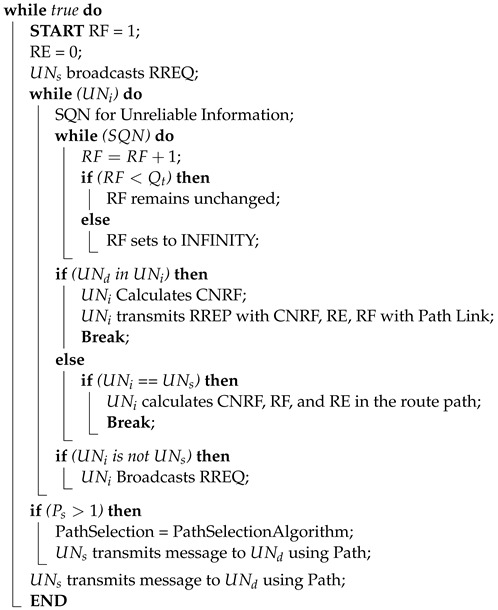


### 4.4. Cluster Formation

Based on the availability of spectrum, our proposed clustering mechanism strategically organizes the network into distinct logical groups. After completing the neighbor discovery phase, each node compiles its list of neighbors and exchanges Accessible Channels Lists (ACLs) with neighboring nodes within a single hop. Our clustering scheme is designed to minimize the number of clusters while simultaneously maximizing the common channels within each cluster.

The process of forming clusters, as outlined in Algorithm 2, is framed as a maximum edge biclique problem. The objective is to create clusters that encompass the greatest number of nodes while incorporating the maximum possible number of shared channels within each cluster, as illustrated in Figure 4.
**Algorithm 2:** Cluster Formation
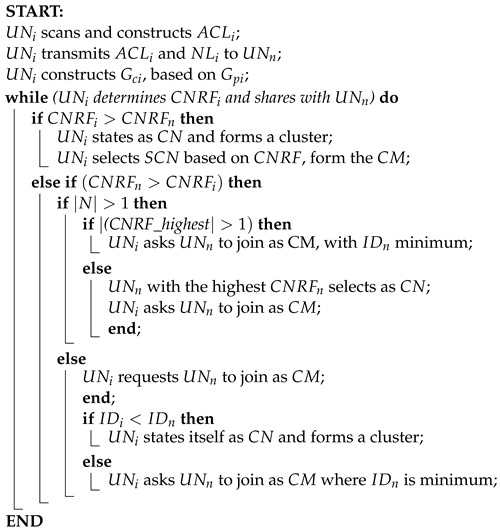


Initially based on nodes and channel list, the maximum bipartite vertices set can be split into two disjoint sets. where the connected nodes to the ACL’s big set can be split into a weighted graph and maximum biclique are taken into consideration. Using this, all nodes are connected to the available channels, so if one of the channels need to get back to it’s PU and the other nodes can move forward with the other maximum left ACLs. The nodes then forms a cluster using the accessible channel list and the maximum biclique.

Using Equation (12), it set the value for all the nodes, and by the maximum weight value it decides the Central Node (CN) in the cluster. Same as that the second highest valued node becomes the Secondary Central Node (SCN). By following the same process, all the nearby nodes creates their own cluster and cluster heads. Moreover, the node that is connected with two or more clusters CN is nominated as Nearby Node (NN). which works as a bridge into two nearby clusters while communication and optimizes the energy and makes the communication faster.

From the proposed Algorithm 2, the Initial UAV Node ($UAV_{i}$) scans and constructs the initial Available Channel List ($ACL_{i}$). Then the $UN_{i}$ transmits $ALC_{i}$ and initial Neighbor List $NL_{i}$ to the neighboring UAV nodes $UN_{n}$. $UN_{i}$ then generates the initial Maximum edge biclique graph ($G_{ci}$) from the initial bipartite graph ($G_{pi}$). $UN_{i}$ then determines the value of the initial Central Node Resolution Factor ($CNRF_{i}$) and shares with the $UN_{n}$.

If the value of $CNRF_{i}$ is greater than the value of the Neighboring Central Node Resolution Factor ($CNRF_{n}$), the $UN_{i}$ node states as the Central Node (CN) and forms its cluster. Moreover, it selects the Secondary Central Node (SCN) based on the second highest value of $CNRF_{i}$ among the Cluster Members (CM).

If the $CNRF_{n}$ is greater than the $CNRF_{i}$, then it check if the number of channels is more than 1 or not. Again, if the CNRF’s highest value is more than 1, then the initial node asks the CN to join as CM where the ID of the neighboring node is minimum. Else, the $UN_{n}$ states as CN directly with its ID and $UN_{i}$ asks $UN_{n}$ to join the cluster as CM.

If the Channel Number (N) is less than 1. the $UN_{i}$ requests to join $UN_{n}$ as CM. If the initial ID ($ID_{i}$) is less than the neighboring ID ($ID_{n}$) then $UN_{i}$ states as CN and form a cluster. Else it requests $UN_{n}$ to join as CM where the $ID_{n}$ is minimum.

In our cluster formation approach, each group is led by a central node (CN). This CN acts as a hub for all communication within the group, so other nodes, called Cluster Members (CMs), rely on it to send and receive messages. It organises clusters in a way that maximizes communication efficiency.

Once a cluster is formed, the CN decides on the best frequencies for communication within the group. If neighboring clusters are detected, the CN assigns some CMs as Nearest Nodes for forwarding nodes (NNs) to improve connectivity with those clusters. NNs help relay messages between different clusters, enhancing overall network performance.

Clusters typically consist of a CN, a Secondary Central Node (SCN), and CMs. However, since the network environment can change rapidly, the proposed protocol is to handle nodes joining or leaving clusters. These protocols ensure that our network remains stable and efficient, even as it adapts to dynamic conditions.

#### 4.4.1. Node Move-In

The Node Move-In protocol outlines how a node can join an existing cluster or form a new one in our proposed clustering mechanism. When a node, denoted as $UN_{j}$, detects a beacon from a neighboring Central Node (CN), it initiates the joining process. Here is how it aligns with our clustering approach:

**Node Identification and Preparation:** The joining UAV node ($UN_{j}$) identifies free channels and prepares its Accessible Channels List (ACL). It then scans for beacons on these channels, allowing it to identify neighboring clusters and nodes.

**Broadcasting and Sharing:** Upon detecting neighboring clusters, $UN_{j}$ broadcasts its neighbor and channel lists. This information exchange facilitates cluster formation and connectivity assessment.

**Graph Construction and CNRF Calculation:** $UN_{j}$ constructs bipartite and maximum edge biclique graphs based on the received information. It then calculates the Central Node Resolution Factor (CNRF) to evaluate connectivity.

**Cluster Joining:** If neighboring CNs are detected, $UN_{j}$ attempts to join the cluster. It selects the CN with the highest CNRF value or the lowest ID if CNRF values are identical. $UN_{j}$ then compares its CNRF with the selected CN and requests to join as a Cluster Member (CM).

**Cluster Formation:** If $UN_{j}$’s CNRF is higher than the selected CN’s, and its neighboring node set matches the CN’s CM list, $UN_{j}$ becomes the new CN, and the existing CN becomes the Secondary CN (SCN). Otherwise, $UN_{j}$ joins as a CM or attempts to form a new cluster if no suitable cluster is found.

**Further Considerations:** If $UN_{j}$ provides better connectivity than the existing Nearest Node (NN) with a neighboring cluster, it becomes the NN. Additionally, $UN_{j}$ can declare itself as CN and form a new cluster if joining existing clusters fails.

This protocol ensures flawless integration of nodes into clusters, optimizing network organization and communication efficiency.

From Algorithm 3, The joining UAV Node ($UN_{j}$) starts to scan its span and prepares the $ACL_{j}$ for $UN_{n}$. $UN_{j}$ then broadcasts $ACL_{j}$ and $UN_{n}$ and create biclique graph ($G_{ci}$) based on bipartite graph ($G_{pi}$) and calculates $CNRF_{j}$. then it shares the value with neighboring $UN_{s}$. After that, it checks if the joining node belongs to the neighboring node or not, if the value of the joining node’s channel is greater than 1 then it again checks if the joining accessible channel list intersects with the central node’s ACL and is greater than 2. The $UN_{j}$ then joins $CN_{j}$ as CM. Otherwise, $CN_{j}$ rejects the joining request from $UN_{j}$ and tries to join with other CNs. However, If $CN_{j}$ fails to join any cluster it makes its cluster and states CN.
**Algorithm 3:** Node Move-In
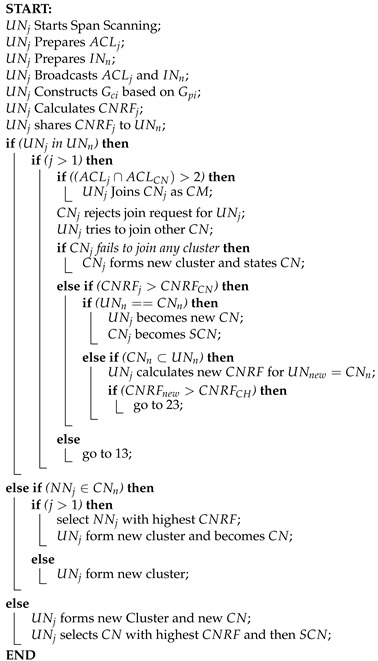


If the value of $CNRF_{j}$ is greater than the value of $CNRF_{C}N$, it checks if $UN_{n}$ is the neighboring CN or not. $UN_{j}$ then becomes the new CN and $CH_{j}$ becomes the Secondary Central Node (SCN). Again if the neighboring cluster head is a proper sub-set of the neighboring UAV Nodes, then the $UN_{j}$ calculates new CNRF for $UN_{new}$ which is the neighboring cluster head ($CN_{n}$). Then it checks if the value of $CNRF_{new}$ is greater than $CNRF_{CN}$ then it follows the same procedure from the algorithm from line 25.

If the $UN_{n}$ is not equal to $CN_{n}$; then it follows the algorithm procedure from line 13. Also, it looks the same if $CNRF_{j}$ is less than $CNRF_{CN}$ and if the channel list is not a maximum biclique edge graph.

When joining the nearest node ($NN_{j}$) belongs to ($UN_{n}$) then it check if the joining node number is more than one, it then selects the $NN_{j}$ with the highest CNRF and $UN_{j}$ forms a new cluster and becomes cluster’s Central node. Otherwise, the result remains the same as before and $UN_{j}$ joins the cluster if the NN does not belong to the set of $UN_{n}$. $UN_{j}$ then joins new cluster with new CN and selects CN with highest CNRF and SCN with second highest CNRF value.

#### 4.4.2. Node Move-Out

The Node Move-Out protocol governs the process of nodes leaving the network, including Central Node (CN), Nearest Nodes (NN), and Cluster Members (CM). Here is how it aligns with our clustering mechanism:

**Leaving Process Initiation:** When a node detects a decrease in beacon signal strength, indicating movement, it initiates the leaving process. Depending on its role as a CN, NN, or CM, different leaving scenarios are considered.

**Leaving Message Broadcast:** The leaving node, denoted as $UN_{l}$, broadcasts a leaving message to its neighbors ($UN_{n}$). $UN_{l}$ first checks its neighbor list to determine its role and potential connections with other nodes.

**CH Departure Handling:** If $UN_{l}$ is a CN, it informs neighboring CMs about the departure and assigns the Secondary CN (SCN) as the new CN. The SCN selects a new CN based on CNRF values, ensuring continuity in cluster leadership.

**NN Departure Handling:** If $UN_{l}$ is an NN, it informs the parent CN ($CN_{k}$) and neighboring clusters about the movement. If direct connection with a neighboring cluster is not possible, $CN_{k}$ requests a CM to act as an NN or selects a new NN based on CNRF values.

**CM Departure Handling:** If $UN_{l}$ is a CM, it informs $CN_{k}$ about the departure. $CN_{k}$ selects a new SCN from the CMs and adjusts intra-cluster communication accordingly. The neighboring cluster is informed about the departure and potential joining scenarios.

**Inter-Cluster Connectivity:** In all leaving scenarios, efforts are made to maintain inter-cluster connectivity. Nodes are assigned new roles or connections based on CNRF values, ensuring network stability and continuity of communication.

This protocol ensures smooth transitions when nodes leave the network, maintaining connectivity and stability within and between clusters.

From Algorithm 4, initially, the leaving UAV ($UAV_{l}$) broadcasts a leaving decision to the neighboring UAVs. Then it checks if the leaving Central Node belongs to neighboring UAVs then it checks if the number of nodes is more than one, and the leaving UAV informs the leaving central node about the leaving information. While the $CN_{l}$ leaves, it checks if the non-leaving members ($CM_{nl}$) do not belong to the leaving central node’s subset and non-leaving cluster members belong to the neighboring central node. If the channel of a non-leaving node is higher than one, the non-leaving cluster members are selected by its CNRF and the leaving central node requests the non-leaving member to be its Nearest Node. The non-leaving members then forward this message to the non-leaving central node ($CN_{nl}$) and the $CM_{nl}$ becomes the NN to maintain the connectivity.
**Algorithm 4:** Node Move-Out
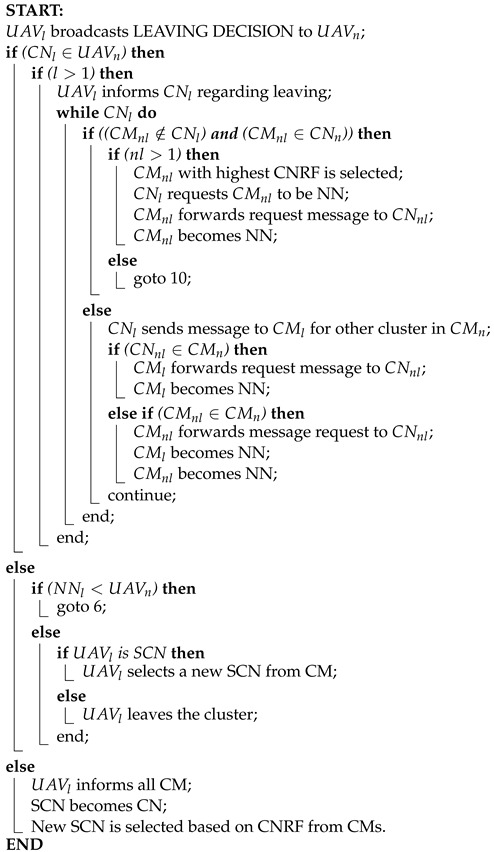


Otherwise, leaving the Central node sends a message to the leaving cluster member for the neighboring member list. if the leaving central node belongs to the neighboring cluster member, then the cluster member forwards the request message to the leaving central node. The leaving member then becomes the Nearest Node (NN). Otherwise, if the non-leaving cluster member forwards a message request to the non-leaving central node the leaving cluster member then becomes the NN along with the non-leaving cluster member for safer connectivity. If the $NN_{l}$ is less than the number of $UAV_{n}$ then it follow the same procedure from line 6.

Otherwise, it checks if the leaving UAV node is the cluster’s secondary Central node or not, if it is then a new SCN is selected from the cluster members, or else the node leaves the cluster.

If the leaving cluster node not belong to the neighboring UAVs, the leaving UAV simply informs all the cluster members and the SCN becomes the new central node and a new SCN is selected based on the CNRF from the CMs.

## 5. Route Selection and Maintenance Process

In cognitive radio ad-hoc networks (CRAHN), routing protocols face unique challenges due to the unpredictable nature of primary user (PU) activity, which influences spectrum availability. Unlike traditional ad-hoc networks, CRAHNs require routing protocols tailored to handle these challenges effectively.

In our network, PUs hold priority in spectrum usage, compelling secondary users (SUs) to swiftly switch channels upon detecting PU transmissions. Additionally, SU radios may need to switch channels during communication due to spatial spectrum variations. Therefore, routing protocols must account for channel switching time, which is the time needed to tune radios to new channels. Moreover, protocols should consider delays caused by queuing and back-off, where back-off delay arises from interference and queuing delay stems from node transmission capacity on specific channels.

A crucial task of cognitive radios is to analyze nearby spectrum and predict PU activity to minimize route interruptions. Consequently, routing protocols must factor in spectrum availability to gauge route stability effectively.

Designing efficient routing protocols for CRAHNs hinges on understanding network objectives. Addressing the aforementioned challenges is vital for creating robust protocols that prioritize stable paths and minimize delays. Our proposed RARE protocol tackles these challenges by selecting suitable nodes as Central Nodes (CNs) and Nearest Nodes (NNs), ensuring maximum route stability. The next section delves into the RARE protocol’s delay-aware routing approach tailored for CRAHNs.

### 5.1. Route Discovery and Selection Process

In this article, the routing protocol is proposed with the route discovery process, where the sending node explores all available routes to the destination node. When a node in our network wants to send packets to another node, it initiates the route discovery by broadcasting a route request message to its immediate neighbors. However, if the sender is a Cluster Member (CM), it directly sends the request to its Central Node (CN), which then broadcasts the request further.

During route discovery, only Central Nodes (CNs) and Nearest Nodes (NNs) are actively involved, while intermediate CMs remain inactive. If a CM receives a route request, it ignores it unless it has the intended destination. CNs or NNs, on the other hand, relay the request to their neighbors until it reaches the destination node.

When a relay node receives the request, it calculates switching, back-off, and queuing delays to determine the total link delay to communicate with the destination. It then forwards a route reply message containing the path delay and node IDs to the neighbor that previously broadcasted the request. This process continues until the sender node receives the reply. Using Equation (10) can be an equation for path Delay ($\partial_{r}^{P}$).
(13)$$\partial_{r}^{P}=\Sigma_{i,jP_{th}\left(s,d\right)}\partial^{D}P_{i,j}$$Upon receiving the reply, the sender calculates the link delay and updates the Path array with the path delay and the complete path from the source to the destination. The sender selects the route with the least delay for message transmission. If multiple routes are available, the sender chooses the one with the lowest delay. If only one route is discovered, it is used for message transmission. This process continues until all possible paths from the sender to the destination are identified and stored in the Path array.

Algorithm 5 describes a method for selecting the optimal route for data transmission between a source UAV and a destination UAV based on real-time measurements. These measurements include Current Node Resolution Factor (CNRF), Routing Efficiency (RE), and Routing Failure (RF) associated with each available path. By using the most up-to-date values of these metrics, the algorithm ensures efficient and reliable data transmission.
**Algorithm 5:** Route Discovery Algorithm
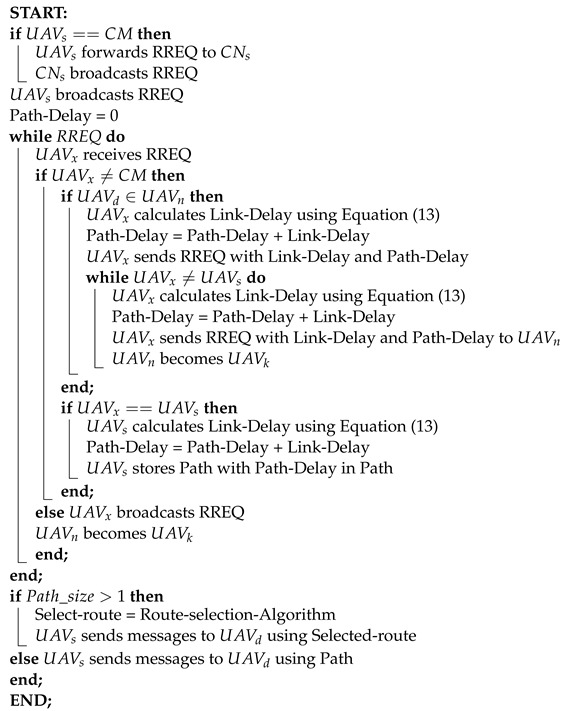


In the event of a link failure during data transmission, the algorithm initiates a route discovery process to identify alternative pathways. Upon detecting a failure, the source UAV receives an error notification indicating the issue, prompting it to update its routing table by removing the faulty node from the list of viable routes. This adaptive mechanism guarantees continuous connectivity and reliable data delivery between the source and destination nodes.

To maintain a robust connection, the source UAV dynamically adjusts its routing strategy, utilizing newly discovered or alternative paths for data transmission to the destination UAV. This adjustment is crucial for mitigating disruptions caused by link failures and ensuring uninterrupted communication in UAV networks.

The algorithm emphasizes the importance of responsiveness to network dynamics, leveraging real-time metrics to optimize routing decisions and promptly addressing failures through efficient route discovery and adaptation processes. This approach ensures that UAV-based communication systems maintain high reliability and performance even in challenging operational environments.

The Route Discovery Algorithm 6 begins by checking if the source UAV ($UAV_{s}$) is a Cluster Member (CM). If it is, $UAV_{s}$ forwards the Route Request (RREQ) to the Control Node ($CN_{s}$) which then broadcasts the RREQ. Otherwise, $UAV_{s}$ broadcasts the RREQ directly and initializes Path-Delay to 0. As RREQ messages are processed, each UAV ($UAV_{x}$) that receives an RREQ and is not a CM checks if the destination UAV ($UAV_{d}$) is within its neighborhood ($UAV_{n}$). If the destination is found, $UAV_{x}$ calculates Link-Delay and updates Path-Delay, then sends the updated RREQ to the next UAV in the path until $UAV_{x}$ reaches $UAV_{s}$ where the path and delay are stored. If the receiving UAV is a CM, it broadcasts the RREQ. After processing all RREQ messages, if multiple paths are available, a Route-selection Algorithm determines the optimal route. The source UAV then sends messages to the destination UAV using the selected or available path. Considering the number of Cluster (n) and the Possible routes (m), the algorithm begins by checking if $UAV_{s}==CM$. The complexity for forwarding the RREQ message is O(n). The broadcasting and receiving operations also have a complexity of O(n). Calculating path delays has a complexity of O(n) as well. Sorting the path for single cluster runs in O(1)=O(n) time. For *m* number of possible routes, the overall complexity of the algorithm is O(logm∗n).
**Algorithm 6:** Route Selection Algorithm
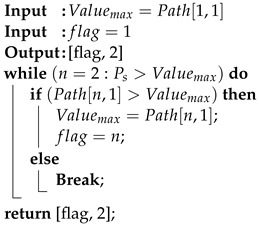


### 5.2. Route Maintainance Process

In our proposed route maintenance Algorithm 7, we address two types of disruptions: link failure and destination failure. When a link in the routing path breaks, the node preceding the broken link sends a route error message to the sender node. Upon receiving this message, the sender node removes the affected route entry from the Path array. It then eliminates all routes containing the broken link from the Path array. Subsequently, the sender node selects a new routing path using the Route Selection Algorithm (Algorithm 5) and utilizes this new route for message transmission. If no alternative route is found in the Path array, the sender node initiates a route discovery process (Algorithm 4) to identify new routes to the destination node.

In the event of destination failure, which could result from the destination node’s movement or malfunction, the neighboring node of the destination sends a destination error message to the sender node. Upon receiving this message, the sender node begins the route discovery process (Algorithm 4) to find new routes to reach the destination node.
**Algorithm 7:** Route Maintenance Algorithm
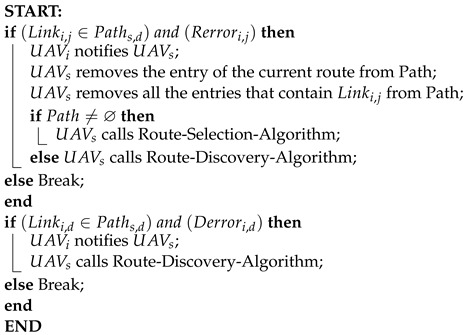


## 6. Simulation Results

The suggested routing protocol’s performance is assessed using the Network Simulators, discrete-event simulator. Each simulation runs for 300 s, and averages are calculated from multiple observations for analysis.

The performance of the suggested routing protocol is evaluated using Network Simulators, a discrete-event simulator, with each simulation running for 300 s and averages calculated from multiple observations to ensure reliable analysis. UAVs have a transmission range of 550 m, a value chosen to represent a typical communication range for effective coverage. They use Constant Bit Rate (CBR) traffic with 512-byte packet sizes, selected for balancing efficient data transfer and manageable packet processing, and a 2 Mbps data transmission rate, reflecting common speeds in UAV networks. The simulation area is set at 800 × 800 m to simulate a moderately sized operational environment, while UAV counts range from 100 to 200 to assess the protocol’s scalability across different network densities. The assessment metrics include throughput, measured as the total size of data packets successfully delivered per unit time, and end-to-end latency, which describes the time taken for data packets to travel from source to destination. These initial values and metrics provide a comprehensive and realistic evaluation of the proposed routing protocol’s performance in UAV network scenarios. Table 1 includes the parameters such as simulation time, UAV transmission range, traffic type, packet size, data transmission rate, simulation area, number of UAVs, and assessment metrics.

### 6.1. Performance Evaluation

The graph in Figure 5 illustrates the relationship between the number of nodes in the network and the time it takes for data to be transmitted, known as end-to-end delay. The analysis considers two different packet sizes: 512 bytes and 1024 bytes, conducted within a simulated environment using UAVs with a transmission range of 550 m.

When utilizing 512-byte packets, the end-to-end delay ranges from 0.23 to 0.318 s, while for 1024-byte packets, it ranges from 0.116 to 0.894 s.

The findings indicate that as the number of nodes increases, the end-to-end delay generally decreases. This phenomenon is attributed to the availability of more pathways for data transmission, resulting in improved efficiency. Interestingly, the use of larger packet sizes, such as 1024 bytes, results in slightly higher end-to-end delays compared to smaller packet sizes. This is due to the increased processing and transmission time required for larger packets.

The graph in Figure 6 depicts the relationship between the number of trials and throughput, representing the quantity of data successfully delivered to the target node per unit time. The analysis examines two different packet sizes: 512 bytes and 1024 bytes, conducted within a simulation environment utilizing UAVs equipped with a transmission range of 550 m.

For packet sizes of 512 bytes, the throughput values range from 21.1858 to 27.883 units (e.g., Mbps or packets per second). This variation in throughput reflects the effectiveness of the TDMA method in allocating time slots for data transmission among multiple UAVs.

Similarly, for packet sizes of 1024 bytes, the throughput values range from 42.4218 to 54.9416 units. The higher throughput values for larger packet sizes indicate the efficient utilization of allocated time slots, allowing for increased data delivery rates per unit time.

The TDMA method plays a crucial role in managing these throughput values by efficiently dividing the transmission time among UAVs, thereby optimizing data transfer rates within the specified transmission range of 550 m. This approach ensures reliable and efficient communication among UAVs in dynamic environments.

The findings suggest that as the number of trials increases, the throughput tends to fluctuate. This fluctuation may be attributed to various factors such as network congestion, packet loss, or environmental interference. Additionally, it is observed that for both packet sizes, the throughput tends to be higher when using larger packets (1024 bytes) compared to smaller packets (512 bytes). This can be attributed to the larger amount of data being transmitted in each packet, leading to higher throughput.

The graph in Figure 7 illustrates the relationship between trials and the packet delivery ratio, representing the percentage of successfully delivered packets out of the total packets sent. The analysis evaluates two different packet sizes: 512 bytes and 1024 bytes, within a simulation environment utilizing UAVs with a transmission range of 550 m.

For packet sizes of 512 bytes, the packet delivery ratio ranges from 75.454% to 98.456%, while for packet sizes of 1024 bytes, it varies from 72.12% to 97.818%.

The findings indicate that as the number of trials increases, the packet delivery ratio tends to fluctuate. This fluctuation may be attributed to various factors such as network congestion, packet loss, or environmental interference. Additionally, it is observed that the packet delivery ratio remains relatively consistent across different packet sizes, with slight variations within the range of results. This suggests that the proposed protocol effectively maintains packet delivery performance regardless of the packet size being transmitted.

The graph in Figure 8 portrays the relationship between trials and the normalized routing load, which represents the average amount of routing overhead incurred per node for packet transmission. This analysis evaluates two distinct packet sizes: 512 bytes and 1024 bytes, within a simulation environment featuring UAVs with a transmission range of 550 m.

For packet sizes of 512 bytes, the normalized routing load ranges from 5.142 to 18.072, while for packet sizes of 1024 bytes, it varies from 5.646 to 16.426.

The findings suggest that as the number of trials increases, the normalized routing load tends to fluctuate. This fluctuation may stem from changes in network topology, traffic patterns, and packet sizes, all of which influence the routing overhead experienced by individual nodes. Additionally, it is observed that the normalized routing load exhibits different trends for each packet size. For instance, while there is a general increase in routing load with higher trial counts for both packet sizes, the trend is less consistent, with instances of both increase and decrease in routing load.

### 6.2. Performance Comparison

This section offers a comparative analysis of simulation results, evaluating the performance of the proposed protocol against four pre-existing alternatives: MDRMA [25], MA-DP-AoDV [26], MDA-AoDV [26], and SecRIP [17].

According to Figure 9, a common trend is observed across all five routing protocols: end-to-end latency increases as the number of nodes in the network grows. This phenomenon can be explained by the fact that denser networks involve a greater number of nodes participating in the data transmission process. Consequently, this leads to an increase in queuing delays as packets need to be processed by more intermediate nodes before reaching their final destination. Additionally, the likelihood of packet collisions and retransmissions also rises with network density, further contributing to the increased latency.

However, when comparing the performance of the proposed routing protocol with the other four protocols, a significant improvement in end-to-end latency is evident. The proposed routing protocol indicates lower latency, which can be attributed to its innovative approach to route selection. Unlike the other protocols that may focus simply on metrics such as hop count or link quality, the proposed protocol integrates the consideration of queuing delays along with other parameters, such as link stability and residual energy, when selecting the optimal routes. This comprehensive evaluation ensures that the selected paths are not only reliable and energy-efficient but also optimized for minimal delay.

By proactively managing queuing delays and avoiding congested routes, the proposed protocol effectively reduces the overall time taken for data packets to travel from the source to the destination. This results in a more efficient and responsive network, particularly in scenarios with high node density. The improved performance of the proposed protocol highlights its robustness and adaptability in managing the complexities associated with dynamic and dense network environments, thereby ensuring more consistent and lower end-to-end latency compared to traditional routing protocols.

On the other hand, Figure 10 demonstrates that an increase in traffic density leads to a corresponding rise in throughput across all the evaluated routing protocols. Throughput, in this context, refers to the total amount of data successfully delivered to the destination node per unit time. As traffic density increases, more data packets are transmitted within the network, thereby boosting the overall throughput.

Among the protocols examined, the proposed routing protocol exhibits superior performance in terms of network throughput when compared to MDRMA [25], MA-DP-AoDV [26], MDA-AoDV [26], and SecRIP [17]. This enhanced throughput performance can be attributed to several key features of the proposed protocol. Firstly, it selects stable routes that are characterized by higher link stability, even in scenarios where spectrum availability is variable. By focusing on link stability, the proposed protocol ensures that data packets are transmitted over reliable paths, thereby reducing the likelihood of link failures and packet loss.

Moreover, the proposed protocol requires fewer control messages to maintain these stable routes. This efficiency in control message overhead further contributes to higher throughput, as more bandwidth is available for actual data transmission rather than being consumed by control signaling. In contrast, the other evaluated protocols demonstrate less stability in environments with variable spectrum conditions. These protocols are more prone to frequent link failures and subsequent packet losses, which negatively impact their overall throughput.

The superior performance of the proposed routing protocol in terms of throughput underscores its robustness and adaptability in managing dynamic network conditions. By effectively handling changes in spectrum availability and maintaining stable communication links, the proposed protocol minimizes disruptions in data transmission, thereby maximizing throughput. This capability is particularly beneficial in high traffic density scenarios, where maintaining consistent and reliable data flow is crucial for network performance.

According to Figure 11 comparing OLSR+GPSR [21] with the Proposed protocol, overhead increases as the number of nodes in the network grows. This is due to more nodes participating in data transmission, leading to more control messages and routing information exchanges.

The Proposed protocol consistently shows lower overhead than OLSR+GPSR across all node densities. This improvement is likely due to more efficient routing algorithms, reduced control message exchanges, and advanced metrics for route selection. By effectively managing overhead, the Proposed protocol enhances network performance, particularly in dense environments, making it more scalable and responsive compared to OLSR+GPSR.

According to Figure 12 comparing OLSR+GPSR [21] with the Proposed protocol, throughput decreases as the number of nodes in the network increases. This trend is evident in both protocols, but the Proposed protocol consistently achieves higher throughput than OLSR+GPSR.

The Proposed protocol maintains higher throughput due to its efficient routing strategies and better handling of network traffic. Unlike OLSR+GPSR, which may suffer from increased packet collisions and retransmissions as the network grows, the Proposed protocol likely integrates advanced metrics for route selection, optimizing for factors such as link stability and queuing delays. This ensures that data packets are transmitted more reliably and efficiently, even in denser networks.

By effectively managing network congestion and optimizing route selection, the Proposed protocol enhances overall network performance. This results in a more scalable and robust network, capable of maintaining higher throughput compared to OLSR+GPSR, particularly as the number of nodes increases.

According to Figure 13 comparing OLSR+GPSR [21] with the Proposed protocol, the packet delivery ratio remains relatively stable across different numbers of nodes. Both protocols maintain high delivery ratios, but the Proposed protocol consistently achieves slightly better performance.

The stability and higher performance of the Proposed protocol can be attributed to its effective routing strategies and robust handling of network traffic. Unlike OLSR+GPSR, which may experience occasional drops in packet delivery due to network congestion and packet collisions, the Proposed protocol likely uses advanced metrics for route selection. These metrics ensure reliable and efficient data transmission, resulting in a consistently high packet delivery ratio.

By optimizing route selection and effectively managing network resources, the Proposed protocol ensures a high and stable packet delivery ratio even as the network grows denser. This highlights its robustness and reliability, making it a more effective solution compared to OLSR+GPSR in maintaining efficient data transmission across varying network conditions.

According to Figure 14 comparing OLSR+GPSR [21] with the Proposed protocol, the end-to-end delay varies across different numbers of nodes. Both protocols manage network traffic, but the Proposed protocol consistently achieves significantly lower end-to-end delays.

The superior performance of the Proposed protocol can be attributed to its effective routing strategies and robust handling of network traffic. Unlike OLSR+GPSR, which shows a steady increase in end-to-end delay as the number of nodes increases, the Proposed protocol demonstrates a more controlled delay pattern. This suggests that the Proposed protocol likely uses advanced metrics for route selection, ensuring reliable and efficient data transmission, resulting in consistently lower end-to-end delays.

By optimizing route selection and effectively managing network resources, the Proposed protocol ensures lower end-to-end delays even as the network grows denser. This highlights its robustness and reliability, making it a more effective solution compared to OLSR+GPSR in maintaining efficient data transmission across varying network conditions. The Proposed protocol showcases its ability to handle network congestion and packet collisions more efficiently than OLSR+GPSR. The significant reduction in end-to-end delay across different node counts underscores the Proposed protocol’s superior performance, making it a more reliable and efficient choice for network routing.

## 7. Conclusions and Future Implementations

The proposed routing protocol addresses existing limitations by effectively managing challenges related to delays, speed, link stability, network scalability, residual energy, and reliability, resulting in improved energy efficiency and overall reliability. However, further enhancements are necessary to ensure functional efficiency and accommodate diverse application requirements and communication environments. Notably, this protocol achieves stable cluster formation, reduces re-clustering issues, and improves communication efficiency. Through simulations, this article demonstrates superior performance compared to existing protocols, highlighting significant advancements in neighbor discovery algorithms and network optimization. While the current evaluation is based on simulations, this paper acknowledges the importance of validating the protocol in real-world environments and a up-to-date comparison table for significant enhance the clarity and impact. Due to time and budget constraints, hardware-in-the-loop (HIL) testing was not conducted within this study. As part of future work, HIL testing is planned to provide a more realistic evaluation of the protocol’s performance. Additionally, expand the analysis to include smaller and larger packet sizes, aiming to assess their impact on throughput, latency, and overall network efficiency. This will allow for a more comprehensive evaluation of the proposed routing protocol’s scalability and performance across a wider range of data payload conditions, enhancing the applicability and robustness of the findings in diverse UAV communication scenarios. This will enhance the reliability of the findings and further underscore the potential for the protocol to meet evolving communication demands and environmental conditions. These steps are critical for continued research and development aimed at improving the functionality and adaptability of CRAHNs.

## Figures and Tables

**Figure 1 sensors-24-06334-f001:**
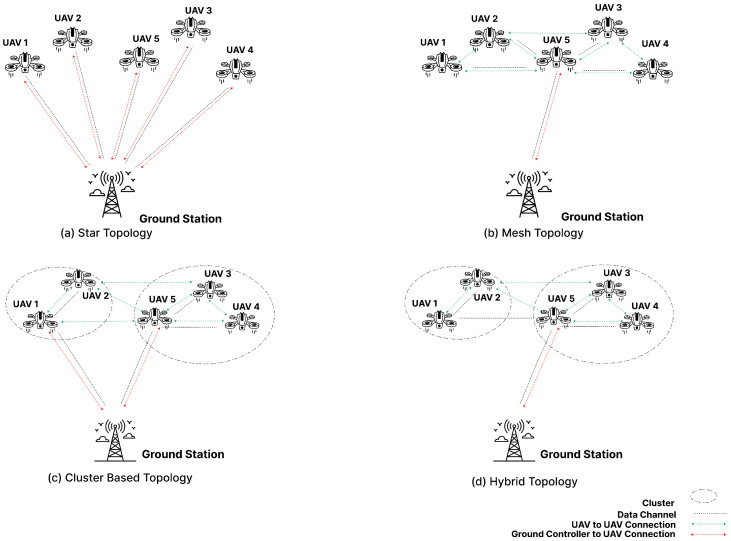
Various Connectivities of UAVs.

**Figure 2 sensors-24-06334-f002:**
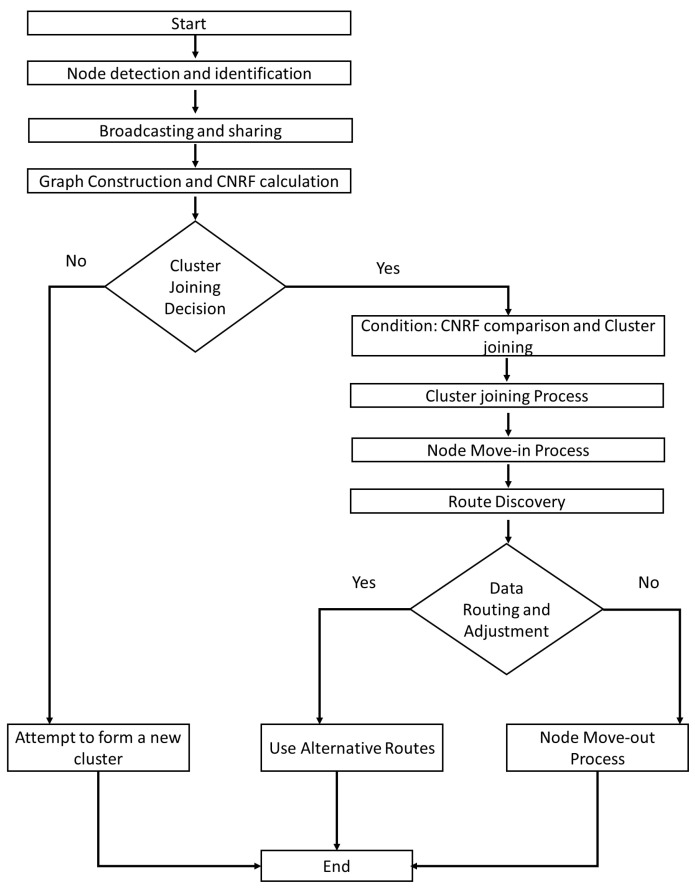
Overview of the Proposed Clustering scheme.

**Figure 3 sensors-24-06334-f003:**
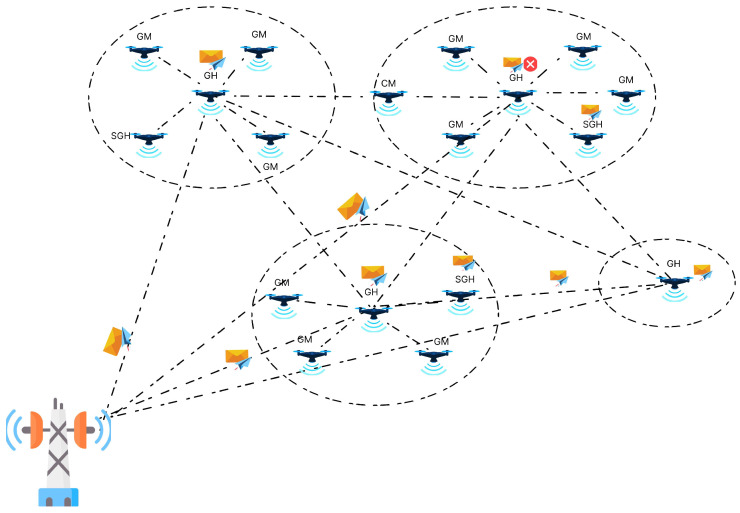
Scenario of communication between GS and CNs.

**Figure 4 sensors-24-06334-f004:**
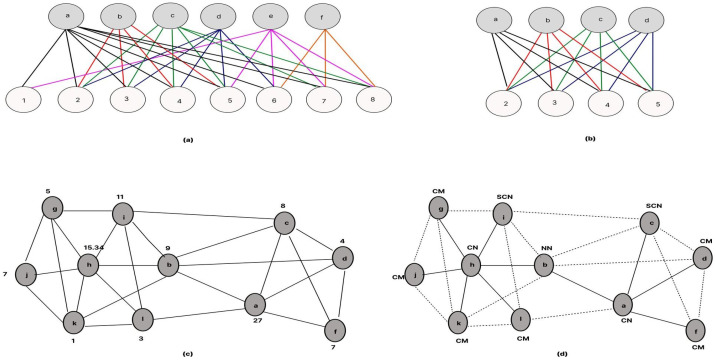
(**a**) Bipartite Graph formation by node $UN_{a}$, (**b**) Maximum edge Biclique graph of node $UN_{a}$, (**c**) Nodes with CNRF value, (**d**) Proposed cluster-based network.

**Figure 5 sensors-24-06334-f005:**
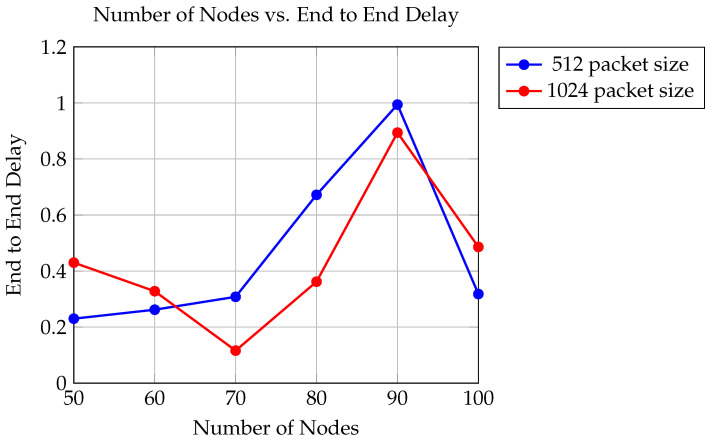
Comparison of End to End Delay for packet sizes of 512 and 1024.

**Figure 6 sensors-24-06334-f006:**
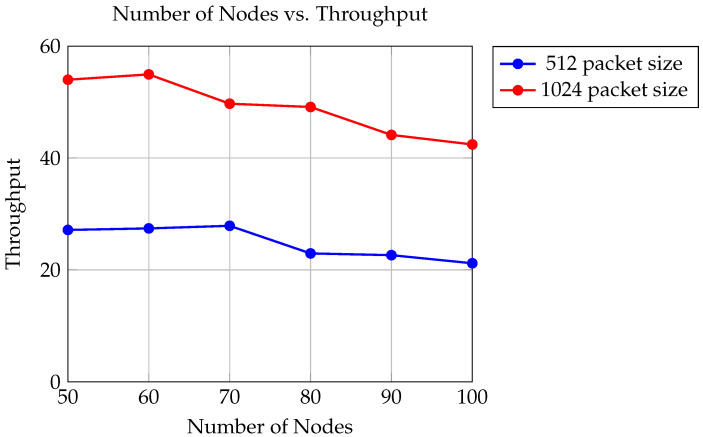
Comparison of throughput for packet sizes of 512 and 1024.

**Figure 7 sensors-24-06334-f007:**
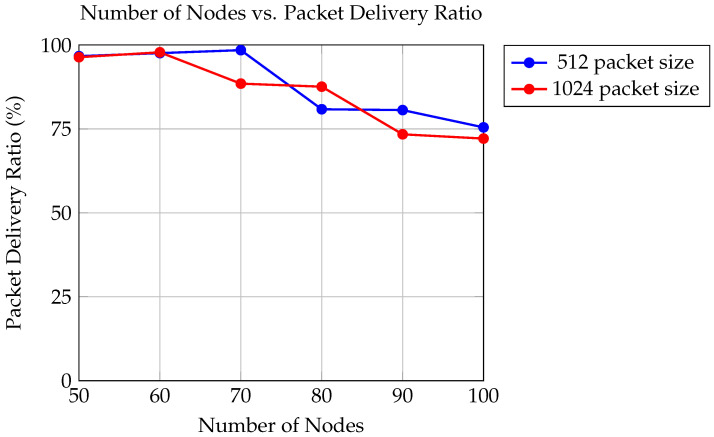
Comparison of packet delivery ratios for packet sizes of 512 and 1024.

**Figure 8 sensors-24-06334-f008:**
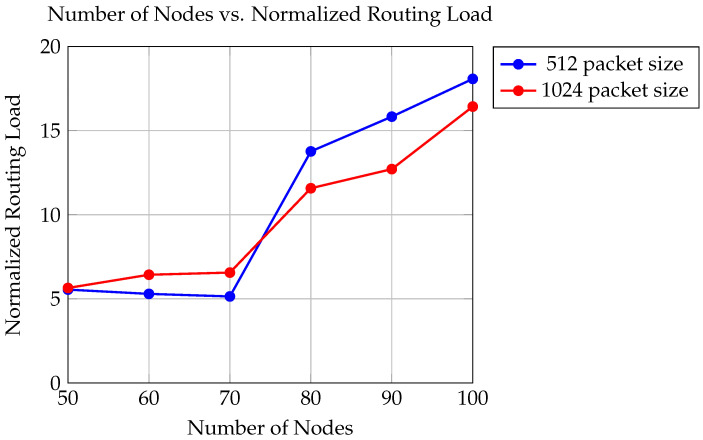
Comparison of normalized routing loads for packet sizes of 512 and 1024.

**Figure 9 sensors-24-06334-f009:**
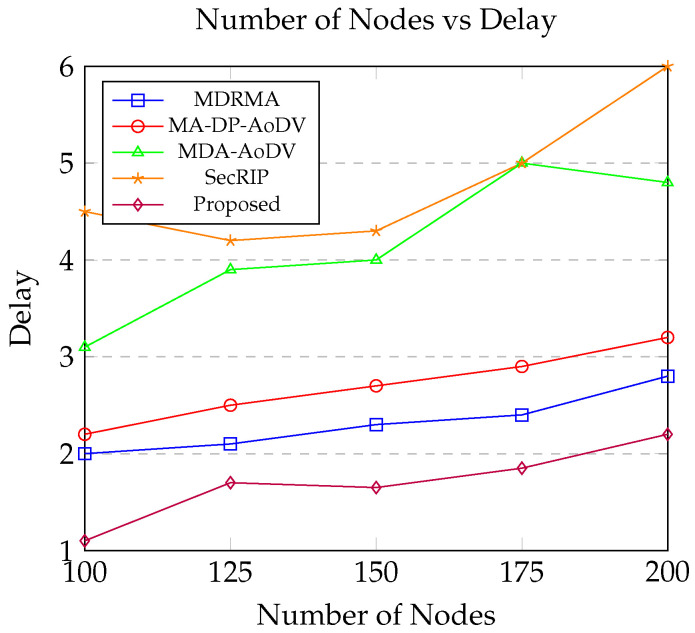
Number of Nodes vs. Delay.

**Figure 10 sensors-24-06334-f010:**
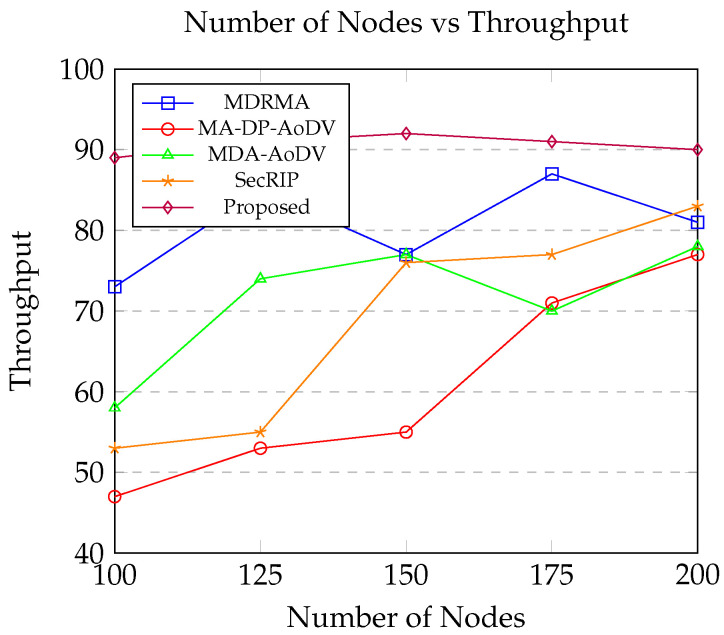
Number of nodes vs. Throughput.

**Figure 11 sensors-24-06334-f011:**
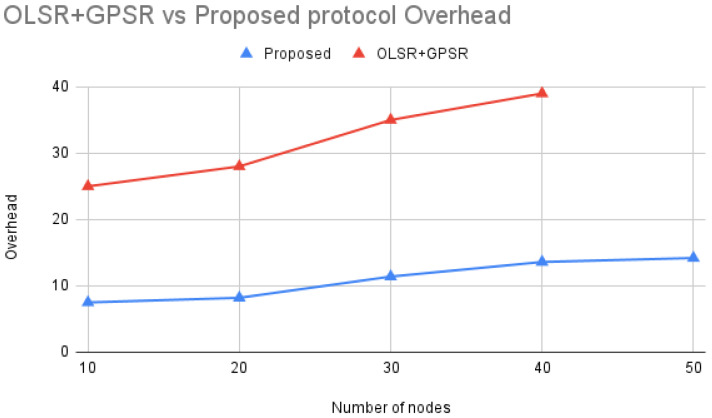
OLSR+GPSR vs. Proposed protocol Overhead.

**Figure 12 sensors-24-06334-f012:**
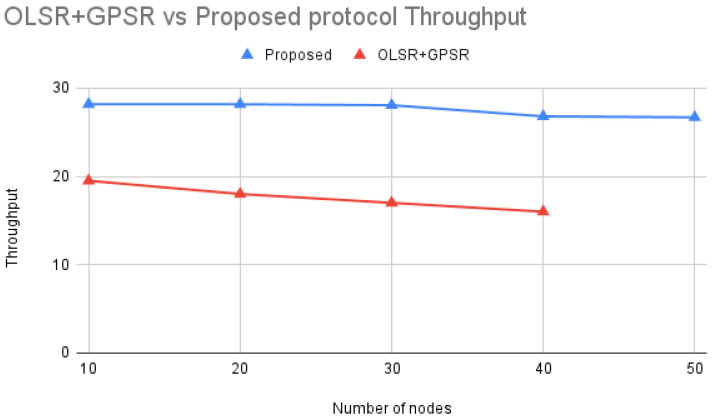
OLSR+GPSR vs. Proposed protocol Throughput.

**Figure 13 sensors-24-06334-f013:**
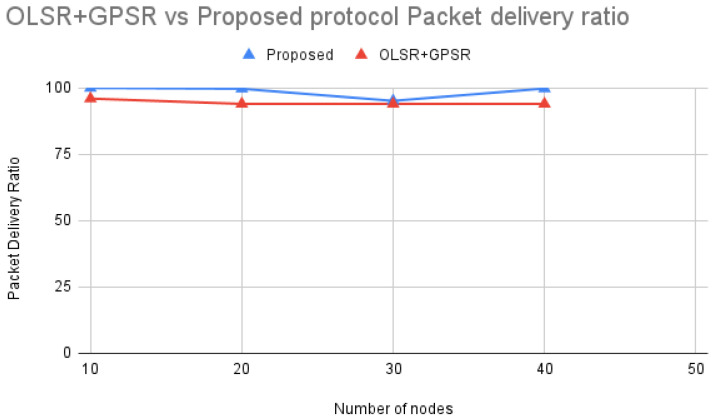
OLSR+GPSR vs. Proposed protocol Packet Delivery Ratio.

**Figure 14 sensors-24-06334-f014:**
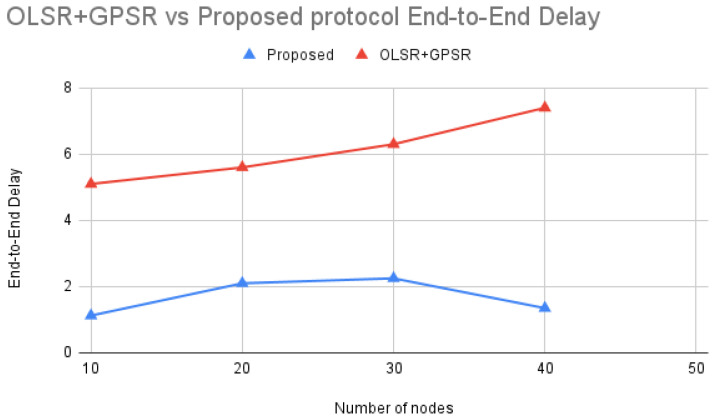
OLSR+GPSR vs. Proposed protocol End-to-End Delay.

**Table 1 sensors-24-06334-t001:** Simulation Parameters.

Parameter	Value
Simulation Time	300 s
UAV Transmission Range	550 m
Traffic Type	Constant Bit Rate (CBR)
Packet Size	512 bytes
Data Transmission Rate	2 Mbps
Simulation Area	800 × 800 m
Number of UAVs	100–200
Assessment Metrics	Throughput, End-to-End Latency

## Data Availability

Data are contained within the article.

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
