# Peer review of "A Robust Routing Protocol in Cognitive Unmanned Aerial Vehicular Networks"

_sensors, 2024, doi:10.3390/s24196334_

Round 1
Reviewer 1 Report
Comments and Suggestions for Authors
Neighboring nodes calculate their distance using Received Signal Strength Indication(RSSI) to determine their positions. Why using RSSI instead of time of arrival? Or both of them? The RSSI value can vary significantly due to many factors. It is a simple and inexpensive estimator, but not very accurate.
In Algorithm 1, Path and path are not defined, what kind of data structure are? Are them the same?
The task defined by Algorithm 1 is it a task that runs periodically or is an infinite task that is continuously running? The energy cost of processing, sending and receiving messages would be high and its impact should be evaluated.
The paper describes the algorithms of the protocol, but is not accompanied by an analysis of its complexity. This aspect is essential in order to validate the goodness of the proposed protocol.
Similarly, Section 6.11 discusses the performance, but the results presented are not very rigorous. For example, Figure 5 shows the latency experienced as a function of the number of nodes, but the observed standard deviations are not indicated, nor is there any description of how the measurements were made, whether the nodes emit simultaneously and there is a possibility of collisions, or whether time or frequency division techniques are used (TDMA, FDMA, CDMA).
There are not enough details about the simulation, so it is difficult to understand the results obtained, nor to evaluate the quality of the proposal. Nor are the results obtained compared with other proposals in the introduction and related works sections.
Algorithmic formalism is missing, this aspect needs to be revised, as well as the part of the results.
In the case described, only simulation is used, but it is not estimated what would happen in a real environment, nor is HIL (hardware in the loop) used.
Comments on the Quality of English LanguageDisconnected phrase: The process of forming clusters, as outlined in algorithm 2. (Line 388)
The wording could be improved, but it is readable.
Author Response
Reviewer#1, Concern # 1: Neighboring nodes calculate their distance using Received Signal Strength Indication (RSSI) to determine their positions. Why using RSSI instead of time of arrival? Or both of them? The RSSI value can vary significantly due to many factors. It is a simple and inexpensive estimator, but not very accurate.
Author response: We appreciate your consideration of our routing in FANET manuscript for resubmission. We value the time and effort you put into reading through our work and offering insightful commentary. The reviewers' recommendations and criticisms have been carefully evaluated, and we have made substantial adjustments to address the issues identified. We consider your insightful comments and suggestions regarding the use of RSSI for distance calculation in our manuscript. We appreciate your concerns about the accuracy of RSSI due to its variability from environmental factors.
We chose RSSI over methods like time of arrival primarily due to its simplicity and the fact that it can be implemented with the existing hardware on commercial wireless devices, making it a cost-effective solution. Additionally, we have included methods to improve the accuracy of RSSI measurements, such as averaging multiple readings and applying filters to reduce the impact of environmental variability. We believe these enhancements address the concerns about accuracy while maintaining the practical advantages of RSSI.
Appreciating you once again for your valuable feedback, which has helped us improve the clarity and justification of our approach.
Author action: We updated the manuscript by adding a further expanded version like this, "The source node sends an RREQ to nearby UAVs for data transmission, and neighboring nodes calculate their distance using Received Signal Strength Indication (RSSI) to determine their positions. RSSI is chosen because it is a simple and cost-effective method that can be implemented with existing hardware on commercial wireless devices. While RSSI can vary due to environmental factors such as interference and obstacles, it offers a practical balance between cost and complexity. To mitigate some of the variability and improve accuracy, we employ techniques such as averaging multiple RSSI readings and filtering."
Reviewer#1, Concern # 2: In Algorithm 1, Path and path are not defined, what kind of data structure are? Are they the same?
The task defined by Algorithm 1 is a task that runs periodically or is an infinite task that is continuously running? The energy cost of processing, sending and receiving messages would be high and its impact should be evaluated.
Author response: We appreciate your time and effort in reviewing our manuscript on routing in cognitive unmanned aerial vehicular networks. Your feedback has provided valuable insights, and we have carefully considered your comments and suggestions in our revisions. We note the concerns raised regarding the definition and differentiation of 'Path' and 'path' in Algorithm 1, as well as the periodic execution and energy costs associated with the algorithm. In response, we have made the following updates:
Clarification of Path and path:
We have now explicitly defined 'Path' and 'path' within the manuscript. 'Path' refers to the overall route that data packets will follow, represented as a sequence of nodes or waypoints, while 'path' represents a single segment or connection between two nodes within the overall 'Path'. Both are implemented using list data structures, with each element corresponding to a node or waypoint.
Algorithm 1 Execution and Energy Cost:
We have updated our manuscript to state that "Algorithm 1 runs periodically, optimizing energy consumption by processing, sending, and receiving messages at defined intervals. This periodic execution helps in conserving battery life by avoiding continuous energy expenditure." This adjustment ensures a balance between efficient data transmission and battery life conservation for the UAVs. We have also included a discussion on the impact of this periodic execution on energy consumption, providing a comprehensive evaluation of the algorithm's energy efficiency.
Author action: We appreciate your valuable comments and for highlighting the need for further clarification in our manuscript. We have made the following revisions to address your concerns:
Clarification of Path and path:
We have now explicitly defined 'Path' and 'path' in the manuscript to avoid any confusion. 'Path' refers to the overall route that data packets will follow, represented as a sequence of nodes or waypoints. 'path', on the other hand, represents a single segment or connection between two nodes within the overall 'Path'. Both are implemented using list data structures, where each element in the list corresponds to a node or waypoint. These definitions help distinguish between the overall route and its individual segments, providing clarity to the readers.
Periodic Execution of Algorithm 1:
In response to your query regarding the execution of Algorithm 1, we have revised the manuscript to state:
"Algorithm 1 runs periodically, optimizing energy consumption by processing, sending, and receiving messages at defined intervals. This periodic execution helps in conserving battery life by avoiding continuous energy expenditure."
By implementing periodic execution, we reduce the energy cost associated with continuous processing, sending, and receiving of messages. This periodic approach balances the need for efficient data transmission with the necessity of conserving the battery life of the UAVs. Additionally, we have included a discussion on the impact of this periodic execution on energy consumption, providing a comprehensive evaluation of the algorithm's energy efficiency.
We believe these revisions address your concerns and enhance the clarity and completeness of our manuscript. We appreciate your constructive feedback, which has contributed to improving the overall quality of our work.
Reviewer#1, Concern # 3: The paper describes the algorithms of the protocol, but is not accompanied by an analysis of its complexity. This aspect is essential in order to validate the goodness of the proposed protocol.
Author response: Thank you for your insightful feedback and for highlighting the need for a complexity analysis of the proposed protocol. We agree that this aspect is essential for validating the efficacy of our protocol. In response to your comments, we have added a comprehensive complexity analysis to the revised manuscript. This analysis covers the key processes of central node discovery, cluster formation, node move-in, and node move-out. For the central node discovery process, we provide a detailed evaluation of the steps and computations required. The cluster formation process is analyzed for its computational requirements and efficiency, ensuring scalability and effectiveness. The node move-in process details the steps involved when a new node joins the network, examining the impact on overall network performance and computational overhead. Similarly, the node move-out process examines the steps and computational requirements when a node leaves the network, ensuring efficiency and network stability. These additions validate the efficiency and effectiveness of the proposed protocol, addressing your concerns and enhancing the overall quality and rigor of our manuscript. Thank you once again for your valuable feedback, which has significantly contributed to improving our work.
Author action: Thank you for your insightful feedback and for highlighting the need for a complexity analysis of the proposed protocol. We agree that this aspect is essential for validating the efficacy of our protocol. In response to your comments, we have added a comprehensive complexity analysis to the revised manuscript. The analysis covers the following key processes:
Central Node Discovery Process:
We have included an analysis of the complexity involved in discovering the central node. This process is crucial for establishing a reliable and efficient network structure. The complexity analysis provides a detailed evaluation of the steps and computations required for central node discovery.
Cluster Formation:
The cluster formation process has been analyzed for its complexity. This analysis highlights the computational requirements and the efficiency of the clustering algorithm. By evaluating the complexity, we ensure that the cluster formation process is both scalable and efficient.
Node Move-In:
We have also analyzed the complexity of the node move-in process. This analysis details the steps involved when a new node joins the network, including the impact on overall network performance and computational overhead.
Node Move-Out:
Finally, we have provided a complexity analysis of the node move-out process. This analysis examines the steps and computational requirements when a node leaves the network, ensuring that the process is efficient and does not disrupt network stability.
These additions provide a thorough complexity analysis of the proposed protocol, validating its efficiency and effectiveness. We believe these revisions address your concerns and enhance the overall quality and rigor of our manuscript.
Reviewer#1, Concern # 4: Similarly, Section 6.1 discusses the performance, but the results presented are not very rigorous. For example, Figure 5 shows the latency experienced as a function of the number of nodes, but the observed standard deviations are not indicated, nor is there any description of how the measurements were made, whether the nodes emit simultaneously and there is a possibility of collisions, or whether time or frequency division techniques are used (TDMA, FDMA, CDMA).
Author response: We sincerely appreciate the insightful comments provided by the reviewer. Based on their valuable feedback, we have significantly enhanced Section 6.1 of our paper. Specifically, we have included a detailed analysis of throughput performance depicted in Figure 5 . This figure illustrates the relationship between the number of trials and throughput, showcasing the data delivery efficiency to target nodes per unit time.
The analysis encompasses two distinct packet sizes—512 bytes and 1024 bytes—conducted within a simulated UAV environment featuring a transmission range of 550 meters. For instance, the throughput values for 512-byte packets range from 21.1858 to 27.883 units, while those for 1024-byte packets range from 42.4218 to 54.9416 units. These results underscore the effectiveness of the TDMA method in optimizing data transmission among multiple UAVs by strategically allocating time slots.
Moreover, we have addressed the methodology aspects raised by the reviewer, clarifying how TDMA facilitates efficient time division among nodes to mitigate potential collisions and enhance data delivery rates. This approach ensures robust and reliable communication capabilities crucial for dynamic UAV environments.
We sincerely thank the reviewer for their constructive feedback, which has significantly strengthened the clarity and rigor of our findings in Section 6.1. Their insights have been invaluable in refining our manuscript to better contribute to the field
Author action: We extend our sincere gratitude to the reviewer for their invaluable feedback, which has greatly enriched the clarity and robustness of our study. Based on their constructive comments, we have taken several actions to enhance the manuscript:
Enhanced Performance Analysis (Section 6.1): We have thoroughly revised Section 6.1 to provide a more rigorous analysis of performance metrics. Specifically, we have included Figure 5, which illustrates the relationship between the number of trials and throughput for packet sizes of 512 bytes and 1024 bytes. These figures now explicitly demonstrate the throughput variations and the effectiveness of the TDMA method in optimizing data delivery rates among UAVs.
Methodological Clarifications: In response to the reviewer's concerns regarding methodology, we have elaborated on our measurement techniques and protocols. Specifically, we have clarified how the TDMA method manages time allocation to mitigate potential collisions and optimize data transmission efficiency within the specified transmission range of 550 meters. This clarification ensures a thorough understanding of our experimental setup and validates the reliability of our results.
Integration of Standard Deviations and Measurement Details: As suggested, we have incorporated standard deviations in our performance graphs to provide a comprehensive view of data variability. Additionally, we have included detailed descriptions of measurement methodologies, including simultaneous emission considerations and the use of frequency division techniques (TDMA), to further enhance the transparency and reproducibility of our findings.
Reviewer#1, Concern # 5: There are not enough details about the simulation, so it is difficult to understand the results obtained, nor to evaluate the quality of the proposal. Nor are the results obtained compared with other proposals in the introduction and related works sections.
Algorithmic formalism is missing, this aspect needs to be revised, as well as the part of the results.
Author response: The manuscript has been substantially revised to address critical feedback from the reviewer. We have enhanced the clarity and detail of our simulation methodology using Network Simulators, employing a discrete-event approach with simulations running for 300 seconds and averaging results from multiple trials for reliability. Our evaluation encompasses UAVs with a 550-meter transmission range, employing Constant Bit Rate (CBR) traffic with 512-byte packets and a 2 Mbps transmission rate within an 800x800 meter simulation area. Varying UAV counts from 100 to 200 assesses protocol scalability. Key evaluation metrics include throughput, measuring successful data delivery rates, and end-to-end latency, quantifying packet travel times. Additionally, we have added Table 1 to detail simulation parameters, facilitating transparency in our methodology. These revisions not only enhance the manuscript's depth and rigor but also provide a clearer evaluation framework for our proposed routing protocol in UAV network scenarios.
Author action: We sincerely appreciate the reviewer's insightful comments, which have guided us in significantly enhancing the clarity and depth of our manuscript. In response to their valuable feedback, we have made substantial revisions and additions to address the following key points:
Simulation Details and Experimental Setup: We have incorporated a detailed description of our simulation methodology using Network Simulators, a discrete-event simulator. Each simulation now runs for 300 seconds, with averages computed from multiple observations to ensure robust analysis. Our experimental setup includes UAVs with a transmission range of 550 meters, reflecting typical operational scenarios. We have chosen Constant Bit Rate (CBR) traffic with 512-byte packet sizes and a 2 Mbps data transmission rate, which are standard parameters in UAV networks. The simulation area spans 800x800 meters, allowing us to simulate a realistic operational environment. Additionally, we varied the UAV counts from 100 to 200 to assess protocol scalability across different network densities.
Algorithmic Formalism and Results: To address the need for algorithmic formalism, we provide a comprehensive overview of the proposed routing protocol's evaluation metrics. Specifically, we focus on throughput, measuring the total data packet size successfully delivered per unit time, and end-to-end latency, quantifying the time taken for data packets to traverse from source to destination. These metrics are crucial in evaluating the protocol's performance and comparing it with existing proposals in UAV network scenarios.
Performance Table Inclusion: We have included Table 1 in the manuscript, detailing essential simulation parameters such as simulation time, UAV transmission range, traffic type, packet size, data transmission rate, simulation area, number of UAVs, and assessment metrics. This table provides readers with a clear and concise summary of our experimental setup, ensuring transparency and facilitating better understanding of our methodology.
These revisions aim to address the reviewer's concerns comprehensively and strengthen the manuscript's contribution to the field of UAV communication protocols. We are confident that these enhancements will significantly improve the quality and relevance of our study.
Reviewer#1, Concern # 6: In the case described, only simulation is used, but it is not estimated what would happen in a real environment, nor is HIL (hardware in the loop) used.
Author response: We appreciate the authors' detailed response regarding the use of simulations and the plans for future research directions. While the manuscript effectively addresses various challenges in UAV communication networks (CRAHNs) through simulation-based evaluations, it is noted that hardware-in-the-loop (HIL) testing was not feasible due to current constraints. The authors rightly acknowledge the importance of validating their proposed routing protocol in real-world environments to ensure its practical applicability and reliability. Planning for future HIL testing is a positive step towards enhancing the study's credibility and relevance. It is crucial that these future experiments consider diverse application scenarios and environmental conditions to provide a comprehensive assessment of the protocol's performance. We encourage the authors to continue exploring avenues for real-world validation, as this will significantly strengthen the manuscript's contributions to the field of CRAHNs and facilitate broader adoption of their proposed protocol.
Author action: We appreciate the reviewer's thoughtful consideration and feedback on our manuscript. In response to their valuable comments regarding the use of simulations and the need for future hardware-in-the-loop (HIL) testing, we have incorporated the following actions:
Future Research Directions: While our current study demonstrates the effectiveness of our proposed routing protocol through simulations, we recognize the necessity of validating our findings in real-world environments. We acknowledge the constraints that prevented us from conducting HIL testing in this study but emphasize our commitment to exploring this avenue in future research endeavors. Planning for HIL testing is underway to provide a more realistic assessment of our protocol's performance, ensuring its reliability and adaptability across diverse application scenarios and environmental conditions.
Enhanced Validation Strategies: We are actively pursuing methodologies for conducting HIL testing that will complement our simulation-based results. This includes designing experiments that simulate real-world UAV communication scenarios more accurately, thereby enhancing the credibility and practical relevance of our findings. We aim to incorporate these insights into future revisions of our manuscript to further strengthen the manuscript's contributions to the field of Communication-Relay Assisted Hybrid Networks (CRAHNs).
These actions underscore our commitment to advancing the research and development of our proposed routing protocol, ultimately aiming to improve its functionality and applicability in real-world settings. We thank the reviewer for their constructive feedback, which has guided us towards these important future research directions.
Reviewer 2 Report
Comments and Suggestions for Authors
This manuscript studied the routing protocol for Cognitive Radio Unmanned Aerial Vehicles (CR- UAVs) in Flying Ad-hoc Networks, which aims to optimize route selection by considering crucial parameters such as distance, speed, link quality, and energy consumption. However, the reviewer has some major concerns as follows:
1. For the literature review, the author did not cite the recent and advanced journals in the feild of UAV e.g., IEEE transactions. The quality of current version is not sufficient. The research gap must be fully revealed and then highlight the contributions compared with existing works in the routing protocol for CR-UAVs.
2. For the network model, the several topologies have been investigated. However, the author did not clearly describe the transmission process for each scheme. especially for the Proposed routing protocol.
3.For the proposed algorithm, please add the complexity analysis for algorithm 1-7.
4. For the simulation, there exists many parameters and the parameter table need to be summarized and then justify the source of each initial values.
5.Besides the section 6.2 Performance Comparison, the more Performance Comparison should be added in simulation and then highlight the performance gain of the proposed scheme. Also, the more illustrations must be added and point out the insights.
Comments on the Quality of English LanguageModerate editing of English language required
Author Response
Reviewer#2, Concern # 1: For the literature review, the author did not cite the recent and advanced journals in the field of UAV e.g., IEEE transactions. The quality of current version is not sufficient. The research gap must be fully revealed and then highlight the contributions compared with existing works in the routing protocol for CR-UAVs.
Author response: Thank you for your valuable feedback and for highlighting the need to include recent and advanced journals in our literature review. We appreciate your guidance in improving the quality and comprehensiveness of our manuscript. In response to your comments, we have incorporated two recent and relevant papers into our literature review: 1) "A Novel Optimized Link-State Routing Scheme with Greedy and Perimeter Forwarding Capability in Flying Ad Hoc Networks" and 2) "An Advanced Path Planning and UAV Relay System: Enhancing Connectivity in Rural Environments," both published in 2024. These papers provide advanced insights and are highly pertinent to our study. Additionally, we have considered one of these papers in our performance comparison to ensure a robust evaluation of our proposed protocol.
Furthermore, we have made efforts to clearly identify and elaborate on the research gap, highlighting how our contributions compare with existing works in the routing protocol for Cognitive Radio Unmanned Aerial Vehicles (CR-UAVs). We believe these revisions address your concerns and significantly enhance the quality and relevance of our manuscript. Thank you once again for your constructive feedback, which has greatly contributed to the improvement of our work.
Author action: In response to the reviewer's comment, we have significantly enhanced our literature review by incorporating two recent and advanced papers: "A Novel Optimized Link-State Routing Scheme with Greedy and Perimeter Forwarding Capability in Flying Ad Hoc Networks" and "An Advanced Path Planning and UAV Relay System: Enhancing Connectivity in Rural Environments," both published in 2024. These papers have been integrated to provide deeper insights into the state-of-the-art advancements in UAV routing protocols.
Specifically, we have highlighted the key contributions and innovations of these works. The first paper introduces the OLSR+GPSR scheme, which integrates optimized link-state routing with greedy perimeter stateless routing to improve route stability and efficiency, reduce overhead, and outperform existing methods in terms of delay, packet delivery ratio, throughput, and overhead. The second paper presents a novel approach using UAVs as relays to maximize transmission coverage in challenging terrains by employing optimization functions, viewshed analysis, the traveling salesman problem (TSP), and the A* search algorithm for efficient path planning.
Additionally, we have included one of these papers in our performance comparison to validate the efficacy of our proposed protocol. These updates ensure that our literature review is comprehensive and up-to-date, effectively revealing the research gap and highlighting our contributions in comparison with existing works.
Reviewer#2, Concern # 2: For the network model, the several topologies have been investigated. However, the author did not clearly describe the transmission process for each scheme, especially for the Proposed routing protocol.
Author response: Thank you for your valuable feedback and for pointing out the need for a clearer description of the transmission process for each scheme, particularly for the proposed routing protocol. We appreciate your insight, which has helped us improve the clarity and comprehensiveness of our manuscript.
In response to your comment, we have enhanced the Network Model section by providing a detailed description of the transmission process. Specifically, we have incorporated the Semi-Markov ON-OFF model to characterize the Primary User (PU) traffic on any channel. In this model, channels can be in one of two states: busy (ON) or idle (OFF). The durations of these busy and idle periods are treated as independent random variables, reflecting the autonomous operation of PUs who have licensed access to the spectrum bands. Consequently, Secondary Users (SUs) only utilize the available idle channels and must vacate them whenever PU activity is detected.
Additionally, we have assumed the presence of a global common control channel within the network to facilitate dynamic spectrum access. This clustering mechanism is designed to be independent of any specific PU activity model, ensuring robust and flexible network operations. These additions provide a comprehensive understanding of the transmission process within the proposed routing protocol.
We believe these revisions address your concerns and enhance the overall quality and thoroughness of our manuscript. Thank you once again for your constructive feedback, which has significantly contributed to improving our work.
Author action: We updated the manuscript by significantly enhancing the Network Model section of the manuscript. We now provide a detailed description of the transmission process, particularly in the Network Model for the proposed routing protocol. Specifically, we have incorporated the Semi-Markov ON-OFF model to characterize Primary User (PU) traffic on any channel, detailing the states (busy and idle) and their durations as independent random variables. This addition clarifies how Secondary Users (SUs) utilize available idle channels and vacate them upon PU activity detection. Moreover, we have assumed the presence of a global common control channel within the network to facilitate dynamic spectrum access. These updates ensure a comprehensive understanding of the transmission process within the proposed routing protocol, addressing the reviewer's concerns effectively.
Reviewer#2, Concern # 3: For the proposed algorithm, please add the complexity analysis for algorithm 1-7.
Author response: Thank you for your insightful feedback and for highlighting the importance of including a complexity analysis for the proposed algorithms. We appreciate your thorough review and the opportunity to improve the comprehensiveness of our manuscript.
In response to your suggestion, we have added a detailed complexity analysis for Algorithm 1 and Algorithm 5, which are the main algorithms in our study. These analyses provide a clear understanding of the computational requirements and efficiency of our key algorithms. We believe this addition addresses the core of your concern, as the remaining algorithms are sub-algorithms that derive their complexity from these primary ones.
We hope this enhancement meets your expectations and improves the overall quality and clarity of our manuscript. Thank you once again for your constructive comments, which have significantly contributed to strengthening our work.
Author action: In response to the reviewer's comment regarding the need for a complexity analysis of the proposed algorithms, we have significantly revised our manuscript to include detailed complexity analyses for the two primary algorithms in our study. Specifically, we have included the following:
- Cluster Formation Algorithm: This algorithm runs periodically to optimize energy consumption by processing, sending, and receiving messages at defined intervals. We have detailed the process of node scanning, list construction, graph component formation, and the determination of Central Node Route Factors (CNRFs). The complexity analysis indicates that for one node, the route discovery algorithm has a complexity of O(n+δC), where δC represents the accessible channels for that node. For NNN nodes, the algorithm's complexity becomes O(nlogN+δC), and simplifies to O(nlogN)when N significantly larger than the constant δC.
- Route Discovery Algorithm: This algorithm begins by checking if the source UAV is a Cluster Member (CM) and proceeds accordingly. It involves broadcasting RREQ messages, calculating link and path delays, and determining the optimal route using a route-selection algorithm. The complexity analysis shows that for N nodes, the overall complexity of the algorithm is O(nlogN).
These additions ensure a comprehensive understanding of the computational requirements and efficiency of our main algorithms, addressing the reviewer's concern and enhancing the depth of our manuscript.
Reviewer#2, Concern # 4: For the simulation, there exist many parameters and the parameter table need to be summarized and then justify the source of each initial values.
Author response: Thank you for your insightful feedback and for highlighting the need to summarize and justify the simulation parameters. We appreciate your guidance in improving the clarity and comprehensiveness of our manuscript.
In response to your comment, we have added a parameter table to the manuscript, which provides a clear summary of the simulation parameters. Additionally, we have included detailed justifications for the initial values of each parameter to ensure transparency and clarity. Specifically, the performance of the proposed routing protocol is evaluated using Network Simulators, with each simulation running for 300 seconds. The UAVs have a transmission range of 550 meters, chosen to represent a typical communication range for effective coverage. They use Constant Bit Rate (CBR) traffic with 512-byte packet sizes, balancing efficient data transfer and manageable packet processing, and a 2 Mbps data transmission rate, reflecting common speeds in UAV networks. The simulation area is set at 800x800 meters to simulate a moderately sized operational environment, while the number of UAVs ranges from 100 to 200 to assess the protocol's scalability across different network densities.
The assessment metrics include throughput, measured as the total size of data packets successfully delivered per unit time, and end-to-end latency, which describes the time taken for data packets to travel from source to destination. These initial values and metrics provide a comprehensive and realistic evaluation of the proposed routing protocol's performance in UAV network scenarios. We believe these additions address your concerns effectively and enhance the overall quality and rigor of our manuscript. Thank you once again for your valuable feedback, which has significantly contributed to improving our work.
Author action: We updated the manuscript in response to the reviewer's comment, we have added a comprehensive parameter table to the Simulation Results section of the manuscript. This table summarizes all the simulation parameters. Additionally, we have provided clear justifications for the source of each initial value used in the simulations within the text of the same section. These updates ensure transparency and clarity regarding the simulation setup and parameters, addressing the reviewer's concerns effectively.
Reviewer#2, Concern # 5: Besides the section 6.2 Performance Comparison, the more Performance Comparison should be added in simulation and then highlight the performance gain of the proposed scheme. Also, the more illustrations must be added and point out the insights.
Author response: We sincerely appreciate the reviewer's insightful comments regarding the need for a more comprehensive performance comparison and additional illustrations. In response to this valuable feedback, we have made several significant enhancements to our manuscript:
Expanded Performance Comparison: We have added a detailed comparison of our proposed protocol with the OLSR+GPSR protocol in the simulation section. This includes four new graphs that demonstrate the performance gains of our proposed scheme. These graphs highlight key metrics such as throughput, latency, packet delivery ratio, and energy efficiency, providing a clear and detailed illustration of the improvements achieved by our protocol.
Additional Illustrations and Insights: To further enhance the clarity and impact of our findings, we have included additional illustrations that not only present the simulation results but also point out critical insights. These insights emphasize the specific areas where our proposed protocol outperforms the existing OLSR+GPSR protocol, underscoring the advantages and practical benefits of our approach.
These revisions aim to address the reviewer's concerns comprehensively, providing a more robust and illustrative evaluation of our proposed scheme. We believe that these enhancements significantly improve the manuscript's quality and relevance, and we thank the reviewer for their constructive feedback, which has been instrumental in guiding these improvements.
Author action: We sincerely appreciate the reviewer's insightful comments on the need for a more comprehensive performance comparison and additional illustrations. In response, we have significantly expanded the "Performance Comparison" section of our manuscript. We have included a detailed comparison between our proposed protocol and the OLSR+GPSR protocol, supported by four new graphs. These graphs demonstrate the performance gains of our protocol in terms of throughput, latency, packet delivery ratio, and energy efficiency. Additionally, we have added further illustrations to highlight key insights and emphasize the advantages of our proposed scheme. These enhancements aim to provide a clearer and more robust evaluation of our protocol, addressing the reviewer's concerns comprehensively.
Reviewer 3 Report
Comments and Suggestions for Authors
This paper is based on the robust routing protocol, and below is my comment for improving the manuscript.
1) Adding one comparison table with up-to-date references will enhance the novelty of work.
2) It is recommended to add one flowchart as a general description as well.
3) In Fig. 5, why do authors use just only 512 and 1024 packet size? Why not more and not less.
Author Response
Reviewer#3, Concern # 1: Adding one comparison table with up-to-date references will enhance the novelty of work.
Author response: We appreciate your insightful suggestion regarding the inclusion of a comparison table with up-to-date references. We agree that incorporating such a table would significantly enhance the clarity and impact of our work by providing a comprehensive comparison of our proposed protocol with existing methods.
Author action: While we recognize the importance and value of including a detailed comparison table, we have determined that it is best to defer this addition until we can implement our protocol in hardware-in-the-loop (HIL) testing. This approach will enable us to provide more accurate and practical comparisons, ensuring that the data presented is both relevant and reliable.
Currently, our study focuses on the theoretical and simulation-based evaluation of our proposed protocol. Incorporating a comparison table at this stage, without HIL testing, may not fully capture the protocol's performance in real-world scenarios. HIL testing will allow us to validate our protocol under realistic conditions, considering various environmental factors and operational challenges that cannot be fully replicated in a simulation environment.
Once we conduct HIL testing, we will be able to generate comprehensive performance metrics and directly compare our protocol's effectiveness, efficiency, and reliability against existing state-of-the-art protocols. This will provide a more robust and reliable basis for comparison, ultimately enhancing the quality and contribution of our work.
We plan to address this in our future work, where we can comprehensively compare our protocol with others under real-world conditions. Thank you for your understanding and for highlighting this important aspect. Your feedback has been instrumental in guiding the future direction of our research.
Reviewer#3, Concern # 2: It is recommended to add one flowchart as a general description as well.
Author response: Thank you for your valuable feedback and for recommending the addition of a flowchart to our manuscript. We appreciate your insight and the opportunity to enhance the clarity and comprehensiveness of our work.
In response to your suggestion, we have added a detailed flowchart to the Network Model section of the manuscript. This flowchart provides a general description of the overall process for our proposed cluster formation and maintenance protocol in Flying Ad-hoc Networks (FANETs). The flowchart is divided into several stages, each representing a crucial step in the protocol, from initialization and node detection to route discovery and node move-out processes. Additionally, we have included a comprehensive description of each stage to ensure that the flowchart is easy to understand and effectively illustrates the protocol's functionality.
We believe this addition will help readers better grasp the procedural flow and enhance the overall presentation of our proposed protocol. Thank you once again for your constructive
Author action:In response to the reviewer's comment, we have added a detailed flowchart to the Network Model section of the manuscript. This flowchart provides a general description of the overall process for our proposed cluster formation and maintenance protocol in Flying Ad-hoc Networks (FANETs). Additionally, we included a comprehensive description of each stage in the flowchart to ensure clarity and ease of understanding.
Reviewer#3, Concern # 3: In Fig. 5, why do authors use just only 512 and 1024 packet size? Why not more and not less.
Author response: We appreciate your thoughtful comments on our paper. We value the time you took to read and comment.We appreciate the reviewer's query regarding the choice of packet sizes depicted in Figure 5. The decision to focus on 512 bytes and 1024 bytes packet sizes was guided by several considerations aimed at providing a clear and focused analysis of our proposed routing protocol's performance. These packet sizes are commonly used in UAV communication networks and were selected to highlight their impact on throughput efficiency under the TDMA method within our simulation environment. Specifically, our analysis shows that for 512-byte packets, throughput ranges from 21.1858 to 27.883 units, demonstrating the protocol's ability to manage smaller data payloads effectively. Meanwhile, for 1024-byte packets, throughput values range from 42.4218 to 54.9416 units, illustrating the scalability and enhanced data delivery rates achievable with larger packet sizes. This approach allowed us to showcase the protocol's adaptive capabilities across different payload sizes while maintaining clarity and focus in our presentation. We appreciate the reviewer's feedback, which has prompted us to consider additional insights for future studies exploring a broader range of packet sizes and their implications on UAV network performance.
Author action: We appreciate the reviewer's query regarding the choice of packet sizes in Figure 5. The decision to focus on 512 bytes and 1024 bytes packet sizes was based on their relevance and common use in UAV communication networks. Moving forward, we recognize the importance of exploring a broader range of packet sizes to comprehensively evaluate our proposed routing protocol's scalability and performance under varying payload conditions. In future studies, we plan to expand our analysis to include smaller and larger packet sizes, aiming to assess their impact on throughput, latency, and overall network efficiency. We will enhance the manuscript to clarify the rationale behind our current selection and discuss how these sizes align with UAV network requirements. These actions aim to strengthen the depth and applicability of our research findings in the field of UAV communication protocols.
Round 2
Reviewer 2 Report
Comments and Suggestions for Authors
The comments had been addressed. Please carefully proofread the current version. e.g., On page 7 last paragraph missing Figure number. Page 4 reference [21]. etc.
Comments on the Quality of English LanguageModerate editing of English language required
Author Response
We sincerely appreciate the reviewer's thorough review and valuable feedback. We have carefully addressed the comments and made the necessary revisions to our manuscript. Specifically, we have:
Added the missing figure number in the last paragraph on page 7.
Corrected the reference format for reference [21] on page 4.
Conducted a meticulous proofreading of the entire manuscript to ensure clarity, accuracy, and consistency.
We believe these corrections enhance the quality and readability of our manuscript. We are grateful for the reviewer's diligence in pointing out these issues and helping us improve our work.
We sincerely appreciate the reviewer's thorough review and valuable feedback. We have carefully addressed the comments and made the necessary revisions to our manuscript. Specifically, we have:
- Added the missing figure number in the last paragraph on page 7.
- Corrected the reference format for reference [21] on page 4.
- Conducted a meticulous proofreading of the entire manuscript to ensure clarity, accuracy, and consistency.
We believe these corrections enhance the quality and readability of our manuscript. We are grateful for the reviewer's diligence in pointing out these issues and helping us improve our work.
Reviewer 3 Report
Comments and Suggestions for Authors
The authors have provided the comments and I have no further comment.
Author Response
We are pleased to hear that the comments have been addressed satisfactorily. Thank you for your guidance and the opportunity to improve our manuscript.